# Sex-structured disease transmission model and control mechanisms for visceral leishmaniasis (VL)

**Temesgen Debas Awoke**[1]*, **Semu Mitiku Kassa**[1], **Kgomotso Suzan Morupisi**[1], **Gizaw Mengistu Tsidu**[2]

**1** Department of Mathematical and Statistical Sciences, Botswana International University of Science and Technology, Palapye, Botswana, **2** Department of Earth and Environmental Sciences, Botswana International University of Science and Technology, Palapye, Botswana

* temesgen.y2000@gmail.com

## Abstract

### Background

Leishmaniasis are a group of diseases caused by more than 20 species of the protozoan that are transmitted through the bite of female sand fly. The disease is endemic to 98 countries of the world. It affects most commonly the poorest of the poor and mainly males. Several research has been conducted to propose disease control strategies. Effective medical care, vector control, environmental hygiene, and personal protection are the mainstays of the current preventative and control methods. The mathematical models for the transmission dynamics of the disease studied so far did not consider the sex-biased burden of the disease into consideration.

### Methodology

Unlike the previous VL works, this study introduces a new deterministic sex-structured model for understanding the transmission dynamics of visceral leishmaniasis. Basic properties of the model including basic reproduction number ($\mathcal{R}_0$), and conditions for the existence of backward bifurcation of the model are explored. Baseline parameter values were estimated after the model was fitted to Ethiopia's VL data. Sensitivity analysis of the model was performed to identify the parameters that significantly impact the disease threshold. Numerical simulations were performed using baseline parameter values, and scenario analysis is performed by changing some of these parameters as appropriate.

### Conclusion

The analysis of the model shows that there is a possibility for a backward bifurcation for $\mathcal{R}_0 < 1$, which means bringing $\mathcal{R}_0$ to less than unity may not be enough to eradicate the disease. Our numerical result shows that the implementation of disease-preventive strategies, as well as effectively treating the affected ones can significantly reduce the disease prevalence if applied for more proportion of the male population. Furthermore, the implementation of vector management strategies also can considerably reduce the total prevalence of the

**Data Availability Statement:** All relevant data are within the manuscript.

**Funding:** This work was carried out with the aid of a grant from the O.R. Tambo Africa Research Chairs Initiative as supported by the Botswana

International University of Science and Technology, the Ministry of Tertiary Education, Science and Technology; the National Research Foundation of South Africa (NRF); the Department of Science and Innovation of South Africa (DSI); the International Development Research Centre of Canada (IDRC); and the Oliver & Adelaide Tambo Foundation (OATF).

**Competing interests:** The authors have declared that no competing interests exist.

disease. However, it is demonstrated that putting more effort in treating affected reservoir animals may not have any significant effect on the overall prevalence of the disease as compared to other possible mechanisms. The numerical simulation infers that a maximum of 60% of extra preventative measures targeted to only male population considerably reduces the total prevalence of VL by 80%. It is also possible to decrease the total prevalence of VL by 69.51% when up to 50% additional infected males receive treatment with full efficacy. Moreover, applying a maximum of 15% additional effort to reduce the number of vectors, decreases the total VL prevalence by 57.71%. Therefore, in order to reduce the disease burden of visceral leishmaniasis, public health officials and concerned stakeholders need to give more emphasis to the proportion of male humans in their intervention strategies.

## Introduction

### Background

Leishmaniasis are a group of diseases caused by more than 20 species of the protozoan genus Leishmania that are transmitted between humans and other mammalian hosts by the vector sand flies [1]. The disease, which affects some of the world's poorest people, is linked to weak immune system, population displacement, substandard housing, malnutrition, and a lack of financial means. In 98 countries of the world, most of which are in the world's poorest regions, an estimated 12 million people have been infected with Leishmania species, and an additional 350 million are at risk of contracting the disease [2]. Clinical manifestations of leishmaniasis include cutaneous leishmaniasis (CL), mucocutaneous leishmaniasis (MCL) and visceral leishmaniasis (VL) [3–5]. Visceral Leishmaniasis is the most severe form of the disease and always fatal if left untreated [6]. Internal organs affected by the disease include liver and spleen. By harming these organs, it also affects the immune system and bone marrow [7]. Spleen and liver enlargement are the disease's defining symptoms. An estimated 50, 000 to 90, 000 new cases of VL occur worldwide annually, with only 25-45% reported to WHO [5]. More than 90% of VL human case occurs in six countries, namely Bangladesh, Brazil, Ethiopia, India, South Sudan, and Sudan. In particular, Eastern Africa has the second-highest number of VL cases [2, 8] with estimated annual incidence of 29, 400–56, 700. The reported VL cases per year in the region is 8, 560 of which Sudan, South Sudan, and Ethiopia have the highest incidence rates, in that order [9]. The disease is endemic in Eritrea, Ethiopia, Kenya, Somalia, Sudan, South Sudan, and Uganda.

In Ethiopia, visceral leishmaniasis (VL) or kala-azar is caused by Leishmania donovani, and is a growing health problem with an estimated annual incidence that ranges between 3700 and 7400 cases with a population at risk of more than 3.2 million [10–12]. The well-known endemic areas of the disease are found in the Metema-Humera lowlands of northwestern Ethiopia bordering Sudan, where it accounts for about 60% of the total cases in arid areas of southern and southwestern parts of the country [13].

VL is a vector-born disease and is transmitted through the bite of female sand fly. The adults are small flying insects of about 2-4 mm in length, with a yellowish hairy body. When the sand fly finishes the early stage of development (eggs, larva, and pupa) and becomes an adult, it finds a host and injects the parasite into the skin during blood meal. This elongated and flagellated parasite grows in the mid-gut of an infected female sand fly and multiplies by the process of simple cell division when reaches into the wound of the host. When a sand fly

ingests these cells from the host cit gets infected with the parasite and the cycle continues [12]. Warm, humid conditions are ideal for sand flies to thrive. They sleep during the day in dark, protected locations like rat tunnels, the bark of old trees, cracks in home walls, and household trash, and they come active at night. Both sexes consume plants, but for the development of their eggs, females additionally require a blood meal. Depending on the species, temperature, and availability of nutrients, the entire life cycle of a sand fly, from egg to adult, can take between 30 and 63 days. The average lifespan of the adult sand fly is 14 days [14].

Numerous public health measures have been implemented to slow the transmission dynamics of VL such as treatment of patients with therapies, reservoir control and vector control. Vectors can be controlled using various techniques, such as personal protection through the use of clothes that cover the whole body when moving into the vector infected areas, residual spraying of dwellings and animal shelters, insecticide-treated bed nets, and the use of windows and door screens to prevent sand flies bites [15]. For the diagnosis of VL, various tests are employed. These include non-leishmanial testing, parasite detection tests, tests for antibodies and tests for antigens. Though these intervention mechanisms being employed, the disease remains to be a public-health threat in many areas of the world.

## Related works

Based on insights from demographic and disease dynamics, mathematical models have been developed and widely used over the past few decades to study a variety of problems in life sciences and medicine (see for e.g [16, 17] and the reference therein). Since Dye [18] created the first mathematical model of VL in 1988, other mathematical models for the disease's transmission dynamics have been developed in an effort to provide policymakers with more accurate predictions of the disease's future burden and potential prevention and control measures.

Along with people, VL also affects dogs and other animals. L. infantum is the causative agent of animal visceral leishmaniasis (ZVL). In order to investigate how to control the spread of VL disease, Zou et al. [19] developed a deterministic mathematical model for visceral leishmaniasis transmission that includes dogs, sand flies, and humans. The study's findings indicate that controlling vertical dog transmission, particularly in asymptomatic dogs, increasing dog vaccination rates, and providing more treatments to infected dogs are the best ways to control the disease. Hussaini et al. [20] developed a mathematical model to analyze the dynamics of animal visceral leishmaniasis (ZVL) transmission, where causative agent is L.infantum in reservoir populations of both human and non-human animal species. Their model includes the non-adult stage of sand flies as well as the human population, reservoir population, and vector population. The analysis of the model is shown to exhibit the phenomena of backward bifurcation. The author fitted the model with Brazilian cases and demographic data and suggested that eliminating sand flies rather than the animal reservoir may be a more efficient strategy for lowering the ZVL burden in the community. It has also been indicated that, controlling infected reservoir has great contribution in reducing transmission. One of the method of controlling infected reservoir is removal of seropositive dogs. However, Rabias et al. [21] suggested that insecticide-impregnated collars, vector, and reservoir control strategies including culling dogs are more effective in reducing human prevalence of ZVL than the destruction of seropositive dogs.

Other studies focused on cost-effective treatments since VL is a disease that affects the poorest of the poor [22], and the disease's treatments are costly and take a long time to the full course. Meheus et al. [23] used a decision tree model to evaluate the cost-effectiveness of all potential VL medicines for the treatment of VL in the Indian subcontinent. The authors demonstrated that the best cost-effective treatment option compared to monotherapies is the combination of liposomal amphotericin B (L-AmB) and miltefosine (MF).

Seasonal fluctuation can have an impact on the dynamics of vector-borne disease transmission and re-emergence. Vector abundance, survival and feeding activity increases with increasing temperature, as well as the rate of development of the pathogen within the vector. Studies that combine preventive and treatments together with seasonal variations have also been conducted. For example, Biswas et al. [24] proposed a SIR-type model for the transmission of VL that considers seasonal variations as a periodic sand fly biting rate. The authors used disease incidence data from South Sudan to validate their model. Their model includes three control strategies: insecticide-treated bed nets; drug-based therapy of Post-kala-azar dermal leishmaniasis (PKDL)-infected and symptomatic patients; and an insecticide spray. The authors concluded that the combination of drug-based treatment for infected patients with an insecticide spray is the most successful, efficient, and cost-effective method of disease control. Mathematical model developed by Gush et al. [15] includes the non-adult stages of sand fly populations as well as the infection's latency periods. They have employed adult vector control, non adult vector control, and treatment of KA patients as their best control measures. Their findings imply that the most cost-effective intervention method in comparison to other strategies is the utilization of treatment for visceral leishmaniasis or kala-azar (KA) patients and raising non adult sand fly mortality rates. Though it is known that, the latency period of the infection plays an important role in the dynamics [25], it was not however taken into account in their investigation.

Evidence shows that biological sex is a variable impacting physiology, immune response, drug metabolism, and consequently, the progression of diseases [2]. For instance, Sjögren syndrome and systemic lupus erythematosus (SLE) affect women more than 80% in the United States [26]. Males are more affected by severe viral, bacterial, and fungal infections than females [27]. Malignant cancer death rates are nearly two times greater in men than in women [28]. In particular, empirical data on leishmaniasis suggest that, men are more frequently infected by Leishmaniasis than women [29–35]. The major reasons that contribute to difference in various endemic areas of the world could be due to social and epidemiological factors [32].

In the literature, researches conducted try to A number of studies have been shown the degree of sex bias as well as identified factors that contributed forwards this difference [31–33]. However, to the best knowledge of the authors, there is no sex dependent mathematical modeling of Viscerial Leishmaniasis transmission in the open literature so far. Therefore, developing model of VL that incorporates sex might provide further insights into dynamics of VL that help understanding the future burden of the disease along sex lines in order to propose the best possible control mechanisms.

## Contribution of this work

Controlling VL is difficult because its distribution is influenced by a complex web of socio-economic, environmental, and demographic factors. However, several studies have been done to provide an insight into efficient control mechanisms. The findings of many past studies have shown that males are highly affected by VL compared to females in different endemic areas of the world such as in Sudan [36–38], in Ethiopia [29, 31, 39], in Indian sub-continent [40], and in Brazil [33]. The males to females ratio is 60% versus 40% in Sudan [36], 77% versus 23% in Kenya [41], and 76% versus 24% in India [42], 87% versus 13% in Ethiopia [43]. To explain this disparity between the sexes, two hypotheses were proposed. The first one is related to social and cultural factors such as the specific behavior, activities, and roles of men and women in society [2, 32]. For example, in Ethiopia more men work in agriculture and/or livestock sectors than women and men wear fewer clothes that expose

their bodies to vector bite than women during the hot evenings and nights [40]. The second factor one is biological differences [40, 44]. Sex is a factor influencing physiology, immunological response, medication metabolism, and ultimately the development of illnesses. Data from experimental models and studies on human infection suggest that there are biological predispositions that contribute to sex-specific parasite load and symptomatic disease after infection with Leishmania species [12, 45, 46]. In addition, access to wealth resources is suggested to be one of the challenges in preventing the spread of infectious diseases. VL mostly affects the poorest of the poor. VL therapy is expensive and takes longer time which adds further economic burden. Thus, these factors should be taken into account in disease control and prevention strategies. Therefore, intervention measures have to take the difference in sex into consideration. The rates of interventions that are applied to the model have to differ across different sexes to identify the most significant model parameters in order to propose an efficient VL control strategy. However, sex-structured VL models were not employed yet that allow different interventions at various rates for men and women population in order to propose efficient disease control strategies. Therefore, in this study a sex structured VL transmission model for human is formulated and studied. Using this model, various intervention mechanisms are simulated and studied.

The paper is organized as follows. A new model for the dynamics of sex-structured VL is formulated in Section 2. Mathematical model analysis are presented in Section 3. Numerical analysis of the model and discussion are carried out in Section 4. Finally, conclusion and future works in Section 5.

## Mathematical model

### The human sub-populations

The total human population ($N_h$) is grouped into two: male human population ($N_m$) and female human population ($N_f$). Then we have classified each group as susceptible male and female human population ($S_m$, $S_f$), exposed men and women human populations ($E_m$, $E_f$) asymptomatically infected male and female human population ($A_m$, $A_f$), KA infected men and women human population ($I_m$, $I_f$) and recovered men and women human population ($R_m$, $R_f$), respectively. At any time t, the total men human population is then given by $N_m = S_m + E_m + A_m + I_m + R_m$ wherecas the total women population is given by $N_f = S_f + E_f + A_f + I_f + R_f$.

Since there is no strong evidence that vertical transmission occurs in VL, we assume that all newborn men and women are fully susceptible. The susceptible human population count is assumed to increase at a constant recruitment rate $\Lambda_m$ and $\Lambda_f$ respectively. Furthermore, upon losing their immunity, recovered people (i.e., the male and female population) become susceptible to the disease and they move to the susceptible male and susceptible female class at a rate of $\psi_1$ and $\psi_2$, respectively. The susceptible human population (men and women) decreases due to the infection from the vector (female sand fly) and due to natural mortality rate $\mu_h$ (we assumed here that the natural mortality rate of male and female humans are equal). Infection on men occur with the force of infection $\lambda_m = c_{1m} \beta_{vh} \frac{I_v}{N_m}$, where $c_{1m}$ is the number of bites at which a male host receives per unit of time. Similarly, infection on women occur with the force of infection $\lambda_f = c_{1f} \beta_{vh} \frac{I_v}{N_f}$, where $c_{1f}$ is the number of bites at which a female host receives per unit of time, $\beta_{vh}$ is the transmission probability of the human host being infected through a single infectious bite, and $I_v$ is the total number of infected sand flies. Hence, the rate of change in the number of susceptible men and women

human populations are given by:

$$\dot{S}_m = \Lambda_m - (\mu_h + \lambda_m)S_m + \psi_1 R_r = \Lambda_m - \left(\mu_h + c_{1m}\beta_{vh}\frac{I_v}{N_m}\right)S_m + \psi_1 R_m,$$

$$\dot{S}_f = \Lambda_f - (\mu_h + \lambda_f)S_f + \psi_h R_r = \Lambda_f - \left(\mu_h + c_{1f}\beta_{vh}\frac{I_v}{N_f}\right)S_f + \psi_2 R_f,$$

where the dot represents the time derivative of the state variable. At the end of the incubation period, it is assumed that a proportion $p$ of men become clinically undetectable (asymptomatic) to VL and join asymptomatic compartment ($A_m$), whereas the remaining $1 - p$ proportions will show the clinical symptoms of the disease and join to symptomatic class($I_m$). Furthermore, males during the incubation period decrease due to natural mortality rate ($\mu_h$). Similarly, a proportion $q$ of women is assumed to become clinically asymptomatic to the disease and join asymptomatic class ($A_f$). The remaining $1 - q$ proportion will show clinical symptoms of the disease and join the symptomatic class ($I_f$). The count of females from this cohort decreases due to natural mortality rate by ($\mu_h$). Then, the rate of change in the number of men and women human population at the incubation stage are given by:

$$\dot{E}_m = \lambda_h S_m - (\mu_h + \eta)E_m = c_1 \beta_{vh}\frac{I_v}{N_h}S_m - (\mu_h + \eta)E_m,$$

$$\dot{E}_f = \lambda_h S_f - (\mu_h + \phi)E_f = c_1 \beta_{vh}\frac{I_v}{N_h}S_f - (\mu_h + \eta)E_f,$$

where $\eta$ represent the inverse of incubation period of human population.

Asymptomatically infected individuals will progress to KA infected compartment ($I_m$ and $I_f$) by the rates $\theta_1$ and $\theta_2$ respectively. Some individuals from this class might recover gradually due to their immune response and will join recovered class at the rates $\omega_1$ and $\omega_2$ for men and women individuals respectively. Individuals from both classes depart due to natural death at a rate of $\mu_h$. Hence, the rate of change in the number of men and women human population from Asymptomatic class are given by:

$$\dot{A}_m = p\eta E_m - (\theta_1 + \omega_1 + \mu_h)A_m,$$

$$\dot{A}_f = q\phi E_f - (\theta_2 + \omega_2 + \mu_h)A_f.$$

Symptomatic VL infected humans get treated and recover at an average rate $\omega_1$ for men and $\omega_2$ for women human population, the count of individuals who develop KA will decrease due to natural mortality at a rate $\mu_h$ and disease induced death rates of $\delta_1$ and $\delta_2$ for men and women human population respectively. Hence, the rate of change in the number of men and women human population from infected compartment with clinical sign are given by:

$$\dot{I}_m = (1 - p)\eta E_m + \theta_1 A_m - (\tau_1 + \delta_1 + \mu_h)I_m,$$

$$\dot{I}_f = (1 - q)\eta E_f + \theta_2 A_f - (\tau_2 + \delta_2 + \mu_h)I_f,$$

where $\tau_1$ and $\tau_2$ are the recovery rate of men and women human population from symptomatic stage respectively.

World Health Organization (WHO) indicates that VL is a treatable disease and individuals can get cured after effective treatments. Hence, men and women individuals who have got VL treatment recover from the disease and then the count in this class also decrease by natural mortality rate ($\mu_h$). The recovered male human population who lost their immunity will join the susceptible class ($S_m$) at a rate $\psi_1$. Similarly, recovered female human population who lost their immunity will join the female susceptible class by a rate $\psi_2$. Therefore, the rate of change in the number of men and women human population from recovered compartment are given by:

$$\dot{R}_m = \omega_1 A_m + \tau_1 I_m - (\mu_h + \psi_1)R_m,$$

$$\dot{R}_f = \omega_2 A_f + \tau_2 I_f - (\mu_h + \psi_2)R_f.$$

## The reservoir sub-populations

The total reservoir population is also classified into three sub-classes: Susceptible reservoir ($S_r$), infected reservoir ($I_r$) and recovered reservoir ($R_r$). The total reservoir population ($N_r$) at any time t is given by $N_r = S_r + I_r + R_r$. Susceptible reservoirs are recruited at a constant rate $\Lambda_r$, and acquire leishmaniasis infection following contacts with infected vector by the force of infection $\lambda_r = c_2 \beta_{vr} \frac{I_v}{N_r}$ where $c_2$ is the number of bites at which a reservoir host receives per unit of time, and $\beta_{vr}$ is the transmission probability of reservoirs. When recovered reservoirs lose their immunity, they become susceptible and join the susceptible reservoir class by the rate $\psi_r$. The count in this class decrease by natural mortality rate ($\mu_r$). Then, the rate of change in the number of susceptible reservoir population is given by:

$$\dot{S}_r = \Lambda_r - (\lambda_r + \mu_r)S_r + \psi_r R_r = \Lambda_r - \left(\mu_r + c_2 \beta_{vr} \frac{I_v}{N_r}\right)S_r + \psi_r R_r.$$

Infected reservoir increases due to newly infected reservoir from the susceptible class by the rate $\lambda_r$ and decreases by natural mortality at a rate $\mu_r$, by disease induced mortality rate $\delta_r$ and due to recovery at a rate $\gamma$. Hence, the rate of change in the number of infected reservoir population is given by:

$$\dot{I}_r = \lambda_r S_r - (\mu_r + \delta_r + \gamma)I_r = c_2 \beta_{vr} \frac{I_v}{N_r} S_r - (\mu_r + \delta_r + \gamma)I_r.$$

When infected reservoirs get an effective treatment being recruited at a rate of $\gamma$, they recover. Dogs become infective to sand flies several months (3-6 months) post-treatment [47]. The recovered individuals lose their immunity by the rate $\psi_r$ and become susceptible to the infection. The number of individuals from this compartment decreases by natural mortality at a rate $\mu_r$. Hence, the rate of change in the number of recovered reservoir population is given by:

$$\dot{R}_r = \gamma I_r - (\mu_r + \psi_r)R_r.$$

## Vector sub-populations

The total vector (female sand fly) population is classified into two sub-classes; Susceptible adult female sand fly ($S_v$) and Infected adult female sand fly ($I_v$). The total vector population

($N_v$) at any time t is given by $N_v = S_v + I_v$. Like that of any vector, the adult vector (sand fly) passes through Egg, Larva, Pupa and Adult stages. But, in this study, to reduce the complexity of the model, we are not considering the non-adult stage of sand fly population as they do not directly contribute to the infection. After transition from the non-adult stage, the newly matured adult female sand flies emerge at the rate $\Lambda_v$ and they are assumed to be susceptible and acquire VL infection following contacts with exposed humans ($E_m$ and $E_f$), asymptomatic humans ($A_m$ and $A_f$), symptomatic humans with clinical sign of VL ($I_m$ and $I_f$) or reservoir infected with leishmaniasis ($I_r$) at an average rate equal to

$$\lambda_v = c_{3m}\beta_{hv}\left(\frac{\kappa_1\zeta A_m + \kappa_2 I_m}{N_m}\right) + c_{3f}\beta_{hv}\left(\frac{\zeta A_f + I_f}{N_f}\right) + c_4\beta_{rv}\frac{I_r}{N_r},$$

where $c_{3m}$ and $c_{3f}$ are the number of times a single vector feeds on a human host per unit of time, $\beta_{hv}$ is the transmission probability of infection from infected human to vectors, $c_4$ is the number of times a single vector feeds on a reservoir host per unit of time, and $\beta_{rv}$ is the transmission probability of infection from infected reservoir to vectors. Since asymptomatic individuals have no awareness about their infection status, they move and mix freely. Whereas infected humans are aware of their infection status and show some restraint from mixing. Hence, the probability of asymptomatic patients getting a bite of the vector is greater than that of symptomatic individuals, by a modification parameter $\zeta$. Due to social and related factors, men are highly vulnerable to the bite of the sand fly. Therefore, the mean number of bites that a female human population receives per day is not equal to that of men. Hence, $\kappa_1$ and $\kappa_2$ are modification parameters that balance this difference. The count of the susceptible vector population is reduced due to natural death at a rate of $\mu_v$. Therefore, the rate of change in the number of susceptible vector (sand fly) population is given by:

$$\dot{S}_v = \Lambda_v - (\lambda_v + \mu_v)S_v.$$

$$\text{or} \quad \dot{S}_v = \Lambda_v - \left(\mu_v + c_{3m}\beta_{hv}\left(\frac{\kappa_1\zeta A_m + \kappa_2 I_m}{N_m}\right) + c_{3f}\beta_{hv}\left(\frac{\zeta A_f + I_f}{N_f}\right) + c_4\beta_{rv}\frac{I_r}{N_r}\right)S_v.$$

VL infected vectors assumed not to suffer death from the disease but decreases their number due to natural mortality by the rate ($\mu_v$) and that compartment increases due to newly infected female sand flies from the susceptible compartment by the rate $\lambda_v$. Therefore, the rate of change in the number of infected vector (sand fly) population is given by:

$$\dot{I}_v = \lambda_v S_v - \mu_v I_v$$

$$\text{or} \quad \dot{I}_v = \left(c_{3m}\beta_{hv}\left(\frac{\kappa_1\zeta A_m + \kappa_2 I_m}{N_m}\right) + c_{3f}\beta_{hv}c\left(\frac{\zeta A_f + I_f}{N_f}\right) + c_4\beta_{rv}\frac{I_r}{N_r}\right)S_v - \mu_v I_v.$$

Combining the models for human sub-population, reservoir sub-population, and vector sub-population we obtain the model for Viseral Leishmaniasis transmission dynamics given by the following deterministic system of non-linear differential equations (see Fig 1 for a flow

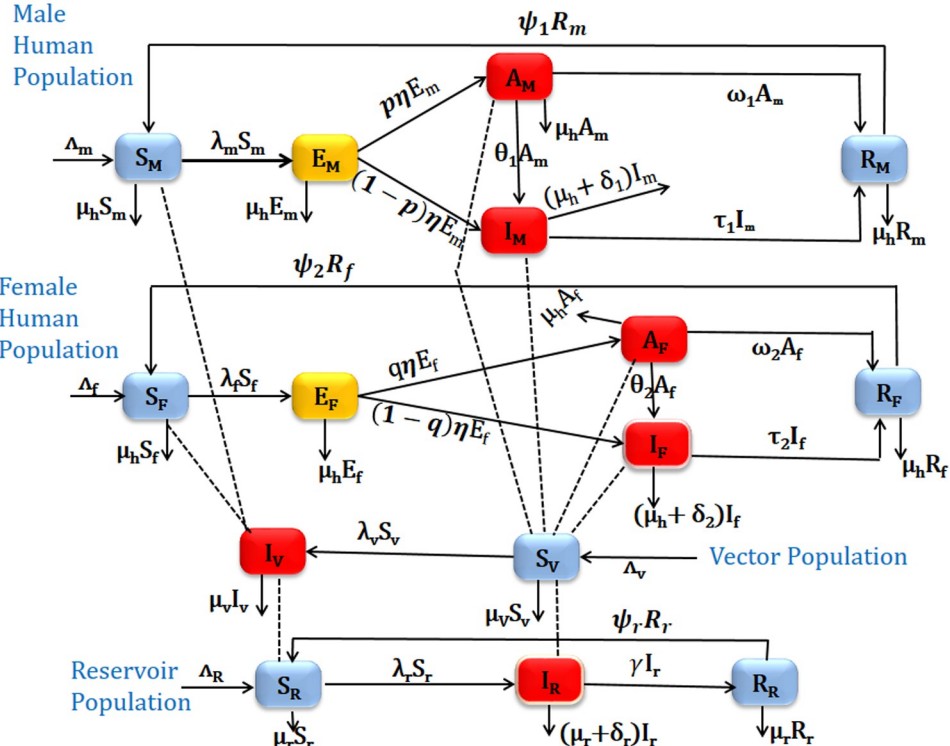

**Fig 1. Schematic diagram for VL compartmental model in which the human population is classified based on their sex.**

diagram and Table 1 for the description of the variables and parameters of the model).

$$
\begin{cases}
\dot{S}_m = & \Lambda_m - (\lambda_m + \mu_h)S_m + \psi_1 R_m \\
\dot{E}_m = & \lambda_m S_m - (\mu_h + \eta)E_m \\
\dot{A}_m = & p\eta E_m - (\theta_1 + \omega_1 + \mu_h)A_m \\
\dot{I}_m = & (1-p)\eta E_m + \theta_1 A_m - (\tau_1 + \delta_1 + \mu_h)I_m \\
\dot{R}_m = & \omega_1 A_m + \tau_1 I_m - (\mu_h + \psi_1)R_m \\
\dot{S}_f = & \Lambda_f - (\lambda_f + \mu_h)S_f + \psi_2 R_f \\
\dot{E}_f = & \lambda_f S_f - (\mu_h + \eta)E_f \\
\dot{A}_f = & q\eta E_f - (\theta_2 + \omega_2 + \mu_h)A_f \\
\dot{I}_f = & (1-q)\eta E_f + \theta_2 A_f - (\tau_2 + \delta_2 + \mu_h)I_f \\
\dot{R}_f = & \omega_2 A_f + \tau_2 I_f - (\mu_h + \psi_2)R_f \\
\dot{S}_v = & \Lambda_v - (\lambda_v + \mu_v)S_v \\
\dot{I}_v = & \lambda_v S_v - \mu_v I_v \\
\dot{S}_r = & \Lambda_r - (\lambda_r + \mu_r)S_r + \psi_r R_r \\
\dot{I}_r = & \lambda_r S_r - (\mu_r + \delta_r + \gamma)I_r \\
\dot{R}_r = & \gamma I_r - (\mu_r + \psi_r)R_r
\end{cases}
\tag{1}
$$

**Table 1. Symbols and description of parameters.**

| Parameters | Description of parameters |
| --- | --- |
| $S_m, S_f, S_v, S_r$ | Susceptible men, women, sand fly, and reservoir population |
| $E_m, E_f$ | Exposed men and women human population |
| $A_m, A_f$ | Asymptomatic male and female human population |
| $I_m, I_f, I_v, I_r$ | Infected men, women, sand fly, and reservoir population |
| $R_m, R_f, R_r$ | Recovered men, women, and reservoir population |
| $\Lambda_m, \Lambda_F$ | recruitment rate of men and women human population |
| $\Lambda_r, \Lambda_v$ | recruitment rate of reservoir (Animal), vector (sand fly) population |
| $1/\mu_h, 1/\mu_v, 1/\mu_r$ | Life expectancy of human, vector(sand fly), reservoir population respectively |
| $\lambda_h, \lambda_v, \lambda_r$ | force of infection for human, sand fly and reservoir population respectively |
| $\theta_1, \theta_2$ | progression rate from asymptomatic to symptomatic |
| $\delta_1, \delta_2, \delta_r$ | disease induced death rates of men, women and reservoir population |
| $\omega_1, \omega_2$ | recovery rate of men and women human population from Asymptomatic stage |
| $\tau_1, \tau_2$ | recovery rate of men and women human population from symptomatic stage |
| $\eta$ | Inverse of incubation period of human population |
| $\psi_1, \psi_2, \psi_r$ | Rate of losing immunity of males, females and reservoirs. |
| $\gamma$ | recovery rate of reservoirs |
| $p$ | the proportion of exposed men who join $A_m$ compartment from $E_m$ class. |
| $q$ | the proportion of exposed women who join $A_f$ compartment from $E_f$ class. |
| $\beta_{vh}, \beta_{vr}$ | The probability that a susceptible humans, and reservoirs becomes infected through a single bite of an infected sand flies respectively |
| $\beta_{hv}, \beta_{rv}$ | The probability that a susceptible sand fly becomes infected when feeding on an infected human, and reservoirs respectively |
| $c_{1m}, c_{1f}, c_2$ | The number of bites in which a male, a female, and a reservoir hosts receives per week |
| $c_{3m}, c_{3f}, c_4$ | The number of times a single vector feeds on a male, a female, and a reservoir hosts |
| $\zeta, \kappa_1, \kappa_2$ | Modification parameters |

$$\text{with} \quad \lambda_m = c_{1m}\beta_{vh}\frac{I_v}{N_m} \tag{2}$$

$$\lambda_f = c_{1f}\beta_{vh}\frac{I_v}{N_f} \tag{3}$$

$$\lambda_v = c_{3m}\beta_{hv}\left(\frac{\kappa_1\zeta A_m + \kappa_2 I_m}{N_m}\right) + c_{3f}\beta_{hv}\left(\frac{\zeta A_f + I_f}{N_f}\right) + c_4\beta_{Rv}\frac{I_r}{N_r} \tag{4}$$

$$\lambda_r = c_2\beta_{vr}\frac{I_v}{N_r} \tag{5}$$

$$N_m = S_m + E_m + A_m + I_m + R_m, N_f = S_f + E_f + A_f + I_f + R_f,$$
$$N_r = S_r + I_r + R_r, N_v = S_v + I_v.$$

The model system (1) is appended with the initial conditions:

$$S_m(0) = S_{m0} > 0, E_m(0) = E_{m0} \geq 0, A_m(0) = A_{m0} \geq 0, I_m(0) = I_{m0} \geq 0,$$
$$R_m(0) = R_{m0} \geq 0, S_f(0) = S_{f0} > 0, E_f(0) = E_{f0} \geq 0, A_f(0) = A_{f0} \geq 0,$$
$$I_f(0) = I_{f0} \geq 0, R_f(0) = R_{f0} \geq 0, S_v(0) = S_{v0} > 0, I_v(0) = I_{v0} \geq 0,$$
$$S_r(0) = S_{r0} > 0, I_r(0) = I_{r0} \geq 0, \ \& \ R_{r0} = R_{r0} \geq 0.$$

(6)

## Model analysis

### Positivity and boundedness of solutions

For the VL model system (1) to be epidemiologically meaningful, it is important to analyze that all its state variables are non-negative for all time. In other words, solutions of the model system (1) with non-negative initial data will remain non-negative for all time $t > 0$. This statement is stated and proved in the following Theorem 1.

**Theorem 1**. [*Positivity*] *Given the initial condition* (6), *the solutions of the model system* (1) *are non-negative for all $t > 0$.*

*Proof.*

Let $S_{m0}, E_{m0}, A_{m0}, I_{m0}, R_{m0}, S_{f0}, E_{f0}, A_{f0}, I_{f0}, R_{f0}, S_{v0}, I_{v0}, S_{r0}, I_{r0}$ and $R_{r0}$ be positive.

We want to show that the state variables are also positive.

From model system (1) $\dot{S}_m = \Lambda_m - (\lambda_h + \mu_h)S_m + \psi_1 R_m$, can be rewritten $\dot{S}_m + (\lambda_h + \mu_h)S_m = \Lambda_m + \psi_1 R_m$. Since $\Lambda_m \geq 0$ & $\psi_1 \geq 0$, it follows that $\dot{S}_m + (\lambda_h + \mu_h)S_m \geq 0$.

After rearranging we have: $\frac{dS_m}{S_m} \geq -(\lambda_h + \mu_h)dt$ and integrating both sides of this expression, we have:

$$\ln(S_m) \geq - \int (\lambda_h + \mu_h)dt + c, \ \text{let} \ \ K(t) = - \int (\lambda_h + \mu_h)dt.$$

Then simplifying the inequality and using the initial value, it follows that

$$S_m(t) \geq S_m(0)e^{K(t)-K(0)} \geq S_m(0) \ \forall t > 0.$$

Since $S_m(0) = S_{m0}$ is positive, that is $S_{m0} > 0$ for all $t > 0$ it implies that $S_m(t) \geq S_m(0) > 0$ for all $t > 0$. We have $S_m(t) > 0$ for all $t > 0$ and we conclude that $S_m(t)$ is non-negative for all $t > 0$. Applying the same procedure for each of the remaining equations, one can argue that all the state variables are positive. That means, $S_m(t), E_m(t), A_m(t), I_m(t), R_m(t), S_f(t), E_f(t), A_f(t), I_f(t), S_v(t), I_v(t), S_r(t), I_r(t), R_r(t)$ are positive $\forall t > 0$.

To describe the boundedness of the solution, we use the following theorem.

**Theorem 2**. [*Invariant region*] *For the model system* (1) *with initial conditions* Eq (6), *the feasible region* $\Omega$, *where* $\Omega = \Omega_1 \times \Omega_2 \times \Omega_3 \times \Omega_4 \times \Omega_5 \subset \mathbb{R}_+^5 \times \mathbb{R}_+^5 \times \mathbb{R}_+^2 \times \mathbb{R}_+^3$ *is positively invariant and attracting with respect to the model system* (1) *for all $t \geq 0$, with,*

$$\Omega_1 = \{(S_m, E_m, A_m, I_m, R_m) : S_m + E_m + A_m + I_m + R_m = N_m \leq \frac{\Lambda_m}{\mu_h}\}$$

$$\Omega_2 = \{(S_f, E_f, A_f, I_f, R_f) : S_f + E_f + A_f + I_f + R_f = N_f \leq \frac{\Lambda_f}{\mu_h}\}$$

$$\Omega_3 = \{(S_v, I_v) : S_v + I_v = N_v \leq \frac{\Lambda_v}{\mu_v}\} \ \&$$

$$\Omega_4 = \{(S_r, I_r, R_r) : S_r + I_r + R_r = N_r \leq \frac{\Lambda_r}{\mu_r}\}.$$

*Proof.* The total men population ($N_m$) is given by $N_m = S_m + E_m + A_m + I_m + R_m$ such that $\dot{N}_m = \dot{S}_m + \dot{E}_m + \dot{A}_m + \dot{I}_m + \dot{R}_m$, and $\dot{N}_m(t) = \Lambda_m - \mu_h N_m(t) - \delta_1 I_m$ implies that $\dot{N}_m(t) - (\Lambda_m - \mu_h N_m(t)) = -\delta_1 I_m$ since $I_m$ is positive, we have

$$\dot{N}_m(t) - (\Lambda_m - \mu_h N_m(t)) \leq \delta_1 I_m. \tag{7}$$

In the absence of VL in the population ($I_m = 0$), then inequality (7) implies that $\dot{N}_m - (\Lambda_m - \mu_h N_m(t)) \leq 0$, hence $\dot{N}_m \leq \Lambda_m - \mu_h N_m(t)$ or $\frac{dN_m(t)}{\Lambda_m - \mu_h N_m(t)} \leq dt$. Integrating from $t = 0$ to $t = T$ gives $\int_0^T \frac{dN_m(t)}{\Lambda_m - \mu_h N_m(t)} \leq \int_0^T dt$. which gives $\ln(\Lambda_m - \mu_h N_m(T)) - \ln(\Lambda_m - \mu_h N_m(0)) \geq -\mu_h T$ or equivalently we get $\frac{\Lambda_m - \mu_h N(T)}{\Lambda_m - \mu_h N(0)} \geq e^{-\mu_h T}$. Further simplification yields, $N_m(T) \leq \frac{\Lambda_m}{\mu_h} - \left(\frac{\Lambda_m - \mu_h N_m(0)}{\mu_h}\right) e^{-\mu_h T}$.

Then by taking the limit of this inequality as $T \to \infty$, we can get $N_m(t) \leq \frac{\Lambda_m}{\mu_h}$, which implies that, $N_m(t) \in \left(N_m(0), \frac{\Lambda_m}{\mu_h}\right)$ for all $t > 0$. Thus $N_m$ is bounded.

In the same way, one can demonstrate that the total women human population $N_f$, the total vector population, $N_v$ and the total reservoir population $N_v$ are bounded.

Therefore, we conclude that the model is epidemiologically feasible and well-posed in $\Omega$.

## Equilibria and stability

The model system (1) has a disease free equilibrium (DFE) given by:

$$\mathcal{E}_0 = \left(\frac{\Lambda_M}{\mu_h}, 0, 0, 0, 0, \frac{\Lambda_f}{\mu_h}, 0, 0, 0, 0, \frac{\Lambda_v}{\mu_v}, 0, \frac{\Lambda_r}{\mu_r}, 0, 0\right). \tag{8}$$

**Basic reproduction number.** As it is indicated in [48], the basic reproduction number ($\mathcal{R}_0$) is defined as the average number of secondary infections that occur when one infective is introduced into a completely susceptible host population. We use the next generation matrix approach proposed by [48] to determine the basic reproduction number, or $\mathcal{R}_0$, of the model system (1), which we obtain as:

$$\mathcal{R}_0 = \sqrt{\mathcal{R}_{0R} + \mathcal{R}_{0M} + \mathcal{R}_{0F}}, \tag{9}$$

where

$$\mathcal{R}_{0R} = \frac{c_2 c_4 \beta_{vr} \beta_{rv} \mu_r \Lambda_v}{\Lambda_r \mu_v^2 (\mu_r + \delta_r + \gamma)},$$

$$\mathcal{R}_{0M} = \frac{\mu_h \Lambda_v \eta c_{1m} c_{3m} \beta_{vh} \beta_{hv}}{\Lambda_m \mu_v^2 (\eta + \mu_h)} \left(\frac{p \kappa_1 \zeta}{(\theta_1 + \omega_1 + \mu_h)} + \frac{p \theta_1 \kappa_2}{(\theta_1 + \omega_1 + \mu_h)(\tau_1 + \delta_1 + \mu_h)} + \frac{(1-p)\kappa_2}{\tau_1 + \delta_1 + \mu_h}\right),$$

$$\mathcal{R}_{0F} = \frac{\mu_h \eta \Lambda_v c_{1f} c_{3f} \beta_{vh} \beta_{hv}}{\Lambda_f \mu_v^2 (\eta + \mu_h)} \left(\frac{(1-q)}{\mu_h + \tau_2 + \delta_2} + \frac{q \zeta}{\mu_h + \theta_2 + \omega_2} + \frac{q \theta_2}{(\tau_2 + \delta_2 + \mu_h)(\theta_2 + \omega_2 + \mu_h)}\right).$$

**Remark 1**. *The formula for $\mathcal{R}_0$ has three parts and is epidemiologically interpreted as follows.*

1. *The quantity $\mathcal{R}_{0R}$ is associated with the infection of susceptible reservoirs by infectious sand flies.*

2. *The quantity $\mathcal{R}_{0M}$ is associated with the infection of susceptible human population (Men) by infectious sand flies.*

3. *The quantity $\mathcal{R}_{0F}$ is associated with the infection of susceptible human population (Women) by infectious sand flies.*

**Theorem 3**. *The Disease free equilibrium (DFE) $\mathcal{E}_0$ of the model system* (1) *is locally asymptotically stable whenever $\mathcal{R}_0 < 1$ and unstable if $\mathcal{R}_0 > 1$.*

*Proof.* This is an immediate consequence of Theorem 2 in [48].

The epidemiological implication of Theorem 3 is that VL can be effectively controlled in the host and reservoir populations (humans and animal reservoirs) if the initial number of infected humans, reservoirs and vector are small enough (i.e., in the basin of attraction of the non-trivial disease-free equilibrium, $\mathcal{E}_0$).

**Endemic equilibrium (EE).** To determine the endemic equilibrium of the model system (1), we equate the right hand side of the model system (1) to zero and obtain the polynomial

$$I_v^* f(I_v^*) = I_v^*[A_3(I_v^*)^3 + A_2(I_v^*)^2 + A_1 I_v^* + A_0] = 0 \tag{10}$$

where

$$A_3 = \mu_v^2 \mu_h^2 a_1^2 a_2 a_3 a_6 a_7 c_{1m} \beta_{hv} c_{1f} \beta_{vh} m_{13} c_2 \beta_{vr}(a_1 a_2 a_3 a_4 - \eta\psi_1(p\omega_1 a_3 + p\tau_1\theta_1 + \tau_1 a_2(1-p)))(a_1 a_6 a_7 a_{10}$$

$$-\eta\psi_2(q\tau_2\theta_2 + q\omega_2 a_7 + \tau_2 a_6(1-q))) + \mu_v a_1^2 a_2 a_3 a_4 a_6 a_7 \mu_h^3 c_{1f} \beta_{vh}^2 m_{13} c_2 \beta_{vr} c_{3m} \beta_{hv} \eta c_{1m}(\zeta p\kappa_1 a_3 + p\kappa_2\theta_1 + \kappa_2 a_2(1-p))$$

$$+\mu_v a_1^2 a_2 a_3 a_6 a_7 a_{10} \mu_h^3 c_{1m} \beta_{hv}^2 m_{13} c_2 \beta_{vr} c_{3f} \beta_{vh} \eta c_{1f}(\zeta q a_7 + q\kappa_2\theta_2 + \kappa_2 a_6(1-q))$$

$$+\mu_v \mu_h^2 a_1^2 a_2 a_3 a_6 a_7 c_{1f} c_{1m} \beta_{vh}^2 \beta_{hv} m_{11} \mu_r c_4 \beta_{rv} c_2 \beta_{vr}(a_1 a_2 a_3 a_4 - \eta\psi_1(p\omega_1 a_3 + p\tau_1\theta_1 + \tau_1 a_2(1-p)))(a_1 a_6 a_7 a_{10}$$

$$-\eta\psi_2(q\theta_2\tau_2 + q\omega_2 a_7 + \tau_2 a_6(1-q)))$$

$$A_2 = \mu_v^2 \mu_h^2 a_1^2 a_2 a_3 a_6 a_7 c_{1m} \beta_{hv} c_{1f} \beta_{vh} \Lambda_r(a_1 a_2 a_3 a_4 - \eta\psi_1(p\omega_1 a_3 + p\tau_1\theta_1 + \tau_1 a_2(1-p)))(a_1 a_6 a_7 a_{10} - \eta\psi_2(q\theta_2\tau_2$$

$$+q\omega_2 a_7 + \tau_2 a_6(1-q))) + \mu_v^2 \mu_h^2 a_1^3 a_2^2 a_3^2 a_4 a_6 a_7 \Lambda_m c_{1f} \beta_{vh} m_{13} c_2 \beta_{vr}(a_1 a_6 a_7 a_{10} - \eta\psi_2(q\theta_2\tau_2 + q\omega_2 a_7 + \tau_2 a_6(1-q)))$$

$$+\mu_v^2 \mu_h^2 a_1^3 a_2 a_6 a_7^2 a_{10} c_{1m} \beta_{hv} \Lambda_f m_{13} c_2 \beta_{vr}(a_1 a_2 a_3 a_4 - \eta\psi_1(p\omega_1 a_3 + p\tau_1\theta_1 + \tau_1 a_2(1-p)))$$

$$+[\mu_h^2 c_{3m} \eta\beta_{hv} c_{1m} \beta_{vh} a_1 a_2 a_3 a_4(p\zeta\kappa_1 a_3 + p\theta_1\kappa_2 + \kappa_2 a_2(1-p))][\mu_h\mu_v a_1^2 a_6^2 a_7^2 a_{10} \Lambda_f m_{13} c_2 \beta_{vr}$$

$$+\mu_v\mu_h a_1 a_6 a_7 \Lambda_r c_{1f} \beta_{vh}(a_1 a_6 a_7 a_{10} - \eta\psi_2(q\theta_2\tau_2 + q\omega_2 a_7 + \tau_2 a_6(1-q)))] + \mu_h\mu_v a_1^2 a_2^2 a_3^2 a_4 \Lambda_m m_{13} c_2 \beta_{vr}$$

$$+\mu_h^2 a_1 a_2 a_3 a_4 c_{3f} \beta_{hv} \eta c_{1f} \beta_{vh}(q\zeta a_7 + q\kappa_2\theta_2 + \kappa_2 a_6(1-q))[\mu_h\mu_v a_1 a_2 a_3 \Lambda_r c_{1m} \beta_{hv}(a_1 a_2 a_3 a_4 - \eta\psi_1(p\omega_1 a_3 + p\tau_1\theta_1$$

$$+\tau_1 a_2(1-p))) - a_1 a_2 a_3 \mu_h \Lambda_v c_{1m} \beta_{hv} m_{13} c_2 \beta_{vr}(a_1 a_2 a_3 a_4 - \eta\psi_1(p\omega_1 a_3 + p\tau_1\theta_1 + \tau_1 a_2(1-p)))]$$

$$+\mu_r\mu_v\mu_h^2 \beta_{rv} \beta_{vr} c_4 m_{11} a_1^3 a_2^2 a_3^2 a_4 a_6 a_7 \Lambda_m c_{1f} \beta_{vh}(a_1 a_6 a_7 a_{10} - \eta\psi_2(q\theta_2\tau_2 + q\omega_2 a_7 + \tau_2 a_6(1-q)))$$

$$-\mu_h^2 m_{11} \mu_r c_4 \beta_{rv} c_2 \beta_{vr} \Lambda_v a_1^2 a_2 a_3 a_6 a_7 c_{1m} c_{1f} \beta_{hv} \beta_{vh}(a_1 a_2 a_3 a_4 - \eta\psi_1(p\omega_1 a_3 + p\tau_1\theta_1 + \tau_1 a_2(1-p)))(a_1 a_6 a_7 a_{10}$$

$$-\eta\psi_2(q\theta_2\tau_2 + q\omega_2 a_7 + \tau_2 a_6(1-q)))$$

$$A_1 = \mu_v^2 \mu_h^2 a_1^4 a_2^2 a_3^2 a_4 a_6 a_7 a_{10} \Lambda_m \Lambda_f m_{13} c_2 \beta_{vr} + a_1^3 a_2 a_3 a_4 a_6^2 a_7^2 a_{10} \Lambda_f \Lambda_r \mu_v \mu_h^3 c_{3m} \beta_{hv} \eta c_{1m} \beta_{vh}(p\zeta\kappa_1 a_3 + p\theta_1\kappa_2 + a_2\kappa_2(1-p))$$

$$+a_1^2 a_2^2 a_3^2 a_4^2 a_5 a_6 a_7 \mu_v \mu_h^3 \Lambda_m \Lambda_r c_{3f} \beta_{hv} \eta c_{1f} \beta_{vh}(q\zeta a_7 + q\kappa_2\theta_2 + a_6(1-q)) + a_1^2 a_2 a_3 a_6 a_7 \mu_v \mu_h^2 m_{11} \mu_r c_4 \beta_{rv} c_2 \beta_{vr}$$

$$-\mu_h^3 \Lambda_v a_1^2 a_2 a_3 a_4 a_6 a_7 c_{3m} \beta_{hv} \eta c_{1m} \beta_{vh}(p\zeta\kappa_1 a_3 + p\kappa_2\theta_1 + a_2(1-p))(a_1 a_6 a_7 a_{10} \Lambda_f m_{13} c_2 \beta_{vr} \Lambda_r c_{1f} \beta_{vh}(a_1 a_6 a_7 a_{10}$$

$$-\eta\psi_2(q\theta_2\tau_2 + q\omega_2 a_7 + \tau_2 a_6(1-q)))) + \mu_v^2 \mu_h^2 a_1^3 a_2 a_3 a_6^2 a_7^2 a_{10} \Lambda_f \Lambda_r c_{1m} \beta_{hv}(a_1 a_2 a_3 a_4 - \eta\psi_1(p\omega_1 a_3 + p\tau_1\theta_1 + \tau_1 a_2(1-p)))$$

$$+\mu_v^2 \mu_h^2 a_1^3 a_2^2 a_3^2 a_4 a_6 a_7 \Lambda_m \Lambda_r c_{1f} \beta_{vh}(a_1 a_6 a_7 a_{10} - \eta\psi_2(q\theta_2\tau_2 + q\omega_2 a_7 + \tau_2 a_6(1-q)))$$

$$A_0 = \mu_v^2 \mu_h^2 \Lambda_m \Lambda_f \Lambda_r a_1^4 a_2^2 a_3^2 a_4 a_6^2 a_7^2 a_{10}(1 - \mathcal{R}_0^2)$$

with $a_1 = \eta + \mu_h, a_2 = \theta_1 + \omega_1 + \mu_h, a_3 = \tau_1 + \delta_1 + \mu_h, a_4 = \psi_h + \mu_h,$
$a_5 = \eta + \mu_h, a_6 = \theta_2 + \omega_2 + \mu_h, a_7 = \tau_2 + \delta_2 + \mu_h, a_8 = \psi_r + \mu_r,$
$a_9 = \mu_r + \delta_r + \gamma, m_{11} = a_8 a_9$ and $m_{13} = a_8 a_9 - \gamma\psi_r$

**Table 2. Number of possible positive roots of $f(I_v^*)$.**

| Cases | $A_3$ | $A_2$ | $A_1$ | $A_0$ | $\mathcal{R}_0$ | No of sign changes | No of possible equilibrium |
|---|---|---|---|---|---|---|---|
| 1 | + | + | + | - | >1 | 1 | 1 |
|   | + | + | + | + | <1 | 0 | 0 |
| 2 | + | + | - | - | >1 | 1 | 1 |
|   | + | + | - | + | <1 | 2 | 0,2 |
| 3 | + | - | + | - | >1 | 3 | 1,3 |
|   | + | - | + | + | <1 | 2 | 0,2 |
| 4 | - | - | + | - | >1 | 2 | 0,2 |
|   | - | + | + | + | <1 | 1 | 1 |
| 5 | - | + | - | - | >1 | 2 | 0,2 |
|   | - | - | + | + | <1 | 1 | 1 |
| 6 | + | - | - | - | >1 | 1 | 1 |
|   | - | + | - | + | <1 | 3 | 1,3 |
| 7 | - | - | - | - | >1 | 0 | 0 |
|   | - | - | - | + | <1 | 1 | 1 |

The solution of Eq (10) are either $I_v^* = 0$ or $f(I_v^*) = 0$. Thus, $I_v^* = 0$ corresponds to the VL disease-free equilibrium point, and the positive solutions of $f(I_v^*) = 0$ corresponds to the endemic equilibrium. When the model associated reproduction number is less than unity, the phenomenon of backward bifurcation is defined by the coexistence of a stable disease-free equilibrium point and a stable endemic equilibrium [49]. By applying Descartes' rule of signs [50] on Eq (10), the various possibilities for the roots of $f(I_v^*)$ are given in Table 2. Based on the existence of the different possible positive roots, we have the following result.

**Theorem 4**. *Given the cases described in* Table 2, *the model system* (1)

1. *has a unique VL persistent equilibrium if cases 1, 2, and 6 are satisfied and $\mathcal{R}_0 > 1$.*

2. *may have more than one VL persistent equilibrium if cases 3, 4, and 5 are satisfied with $\mathcal{R}_0 > 1$.*

3. *may have multiple VL persistent equilibria if cases 2, 3 6 are satisfied with $\mathcal{R}_0 < 1$.*

4. *has no VL persistent equilibrium if case 1 is satisfied and $\mathcal{R}_0 < 1$.*

Theorem 4 (3) shows the co-existence of VL-free equilibrium and VL persistent equilibrium, which indicates that system (1) can undergo a backward bifurcation phenomenon at $\mathcal{R}_0 = 1$. In this case, the VL problem will stay in the population even though the basic reproduction can be reduced to less than unity.

## Numerical analysis of the model

### Parameter estimation

For parameter estimation and numerical simulations, five years (2014-2018) of annual VL confirmed cases in Ethiopia from a research reported in [43] as indicated in Table 3 are used.

The total population in the regional state in 2022, projected based on the 2007 population census, was 22, 876, 991 (Females = 11, 413, 997, Males = 11, 462, 994) with an annual population change of 1.9% [51]. According to the latest WHO data published in 2020 life expectancy in Ethiopia is 66.9 years for male and 70.5 years for female, and the total life expectancy is 68.7 years. Hence, the natural death rate of human population is estimated as, $\mu_h = \frac{1}{68.7 \times 52} =$

**Table 3. Human reported visceral leishmaniasis cases in Amhara regional state, Ethiopia.** Data re-organized from [43].

| Year | Male Cases | Female Cases | Total Cases |
|------|-----------|--------------|-------------|
| 2014 | 783 | 154 | 937 |
| 2015 | 741 | 156 | 897 |
| 2016 | 1059 | 179 | 1237 |
| 2017 | 1160 | 88 | 1248 |
| 2018 | 1129 | 175 | 1304 |
| Total | 4,872 (87%) | 752 (13%) | 5, 624 |

0.0002799 per week. The weekly recruitment rate of males is estimated to be 11, 462, 994 × 1.9%/52 = 4, 188.4. Similarly, the weekly recruitment rate of females is estimated to be 11, 413, 997 × 1.9%/52 = 4, 170.499. Since only a certain section of the population is staying in the VL endemic region, 10% of male and 10% of female weekly recruitment rates are assumed for fitting purpose. The Dog is one of the known reservoir of VL. To determine the natural death rate of reservoirs, the life expectancy of dogs is taken as 10 to 13 years. Hence, $\mu_r = \frac{1}{12 \times 52} = 0.00160$ per week is considered. The average life span of female sand fly is 2 weeks, implying $\mu_v = \frac{1}{2} = 0.5$.

Chapman et al. [25] estimated that the mean duration of asymptomatic infection was 147 days (95% CI 130-166 days) and that of symptomatic was 140 days (95% CI 123-160 days). The mean waiting time in the recovery rate was also stated as 1110 days (95% CI 988-1247 days). The proportion of asymptomatic individuals that develop KA was 14.7% with 95% CI 12.6-20.0%. It is also indicated that, females have a lower rate of progression from asymptomatic infection to recovery with hazard ratio (HR) 0.73 (95% CI 0.57–0.94) and a higher rate of return from recovery to susceptibility with HR 1.36 (95% CI 1.07–1.72). By using these information, we have estimated some of model parameters as follows. For rates of recovery $\omega_1 = 0.027525$, $\omega_2 = 0.020094$, $\tau_1 = 0.028902$, $\tau_2 = 0.021098$, for rate of immunity loss $\psi_1 =$, $\psi_2 =$, and $\theta_1 = 0.147$, $\theta_2 = 0.147$ per week for progression rates. Alvar et al. [47] estimated that dogs lose their immunity and become infective to sand flies 3-6 months after treatment. We took the average of these values. This implies that, $\psi_r = \frac{1}{4.5 \times 4} = 0.055556$ per week. The incubation period (i.e the duration between infection and the first symptom or sign) is generally 3–8 months, but it can be longer or shorter in people with compromised immune systems who visit endemic areas [4]. However, we decided to take the average of these periods. With the above estimates, the initial values are taken to be $S_m(0) = 300, 000$, $E_m(0) = 300$, $A_m(0) = 120$, $I_m(0) = 150$, $R_m(0) = 200$, $S_f(0) = 280, 000$, $E_f(0) = 125$, $A_f(0) = 200$, $I_f(0) = 50$, $R_f(0) = 200$, $S_v(0) = 10, 000$, $I_v(0) = 200$, $S_r(0) = 10, 000$ and $I_r = 200$, $R_r = 150$. To calibrate the model and estimate the parameters from the VL Ethiopia data [43]. MultiStart optimization algorithm is used to obtain a global solution for the objective function (total error) and estimated model parameters. The estimated model parameters are indicated in Table 4.

Fig 2 depicts the female VL infected cases simulated using the parameters estimated from multi-start optimization and observed VL. The model has captured most of the observation points with the exception of one outlier after week 250. The corresponding male VL infected cases are shown in Fig 3.

Fig 4 shows exposed, asymptomatic, symptomatic and recovered male as well as the cases for female counterpart. The number of exposed male increases exponentially until week five and starts decreasing thereafter. The number of recovered male linearly increases with time.

**Table 4. Estimated parameter values and the limits of the 95% percentile bootstrap confidence intervals.** All the 21 parameters out of are within their confidence intervals.

| Parameter | Unit | Range | Parameter value | 95% CI | Source |
|---|---|---|---|---|---|
| $\Lambda_m$ | persons week$^{-1}$ | $N_{m0} \times 1.9\%/52$ | 418.84 | | estimated |
| $\Lambda_f$ | persons week$^{-1}$ | $N_{f0} \times 1.9\%/52$ | 417.269 | | estimated |
| $\mu_h$ | week$^{-1}$ | $1/67.81 \times 52$ | 0.0002799 | | calculated |
| $\mu_r$ | week$^{-1}$ | $1/12 \times 52$ | 0.0016 | | calculated |
| $\mu_v$ | week$^{-1}$ | $1/2$ | 0.5 | | [53] |
| $\theta_1$ | week$^{-1}$ | | 0.147 | | [25] |
| $\theta_2$ | week$^{-1}$ | | 0.147 | | [25] |
| $\omega_1$ | week$^{-1}$ | | 0.027525 | | [25] |
| $\omega_2$ | week$^{-1}$ | | 0.020094 | | [25] |
| $\tau_1$ | week$^{-1}$ | | 0.028902 | | [25] |
| $\tau_2$ | week$^{-1}$ | | 0.021098 | | [25] |
| $\psi_1$ | week$^{-1}$ | | 0.001527 | | [25] |
| $\psi_2$ | week$^{-1}$ | | 0.002077 | | [25] |
| $\psi_r$ | week$^{-1}$ | | 0.055556 | | [47] |
| $\eta$ | week$^{-1}$ | | 0.0625 | | [4] |
| $\Lambda_r$ | reservoirs week$^{-1}$ | (100, 800) | 491.04294 | [104.08235, 784.45252] | fitted |
| $\Lambda_v$ | vectors week$^{-1}$ | (550, 2000) | 691.95421 | [616.02771, 1875.66451] | fitted |
| $p$ | proportion | (0.01, 0.741) | 0.64033 | [0.13894, 0.74095] | fitted |
| $q$ | proportion | (0.7, 0.741) | 0.16619 | [0.07021, 0.54972] | fitted |
| $\beta_{vh}$ | proportion | (0.012, 0.39) | 0.01311 | [0.01200, 0.618337] | fitted |
| $\beta_{vr}$ | proportion | (0.05, 0.39) | 0.15080 | [0.05038, 0.31120] | fitted |
| $\beta_{hv}$ | proportion | (0.05, 0.49) | 0.16093 | [0.05342, 0.48615] | fitted |
| $\beta_{rv}$ | proportion | (0.05, 0.49) | 0.05090 | [0.05021, 0.16306] | fitted |
| $c_{1m}$ | per head week$^{-1}$ | (4, 40) | 23.32835 | [6.55566, 31.57171] | fitted |
| $c_{1f}$ | per head week$^{-1}$ | (1, 32) | 2.83409 | [1.00506, 3.51801] | fitted |
| $c_2$ | per head week$^{-1}$ | (1, 36) | 4.93540 | [1.02156, 18.98113] | fitted |
| $c_{3m}$ | per head week$^{-1}$ | (2, 20) | 18.90792 | [4.01503, 19.89607] | fitted |
| $c_{3f}$ | per head week$^{-1}$ | (2, 20) | 2.74912 | [2.00128, 14.96985] | fitted |
| $c_4$ | per head week$^{-1}$ | (1, 20) | 1.06446 | [1.00021, 3.13139] | fitted |
| $\zeta$ | proportion | (1, 4) | 1.07488 | [1.00702, 3.15381] | fitted |
| $\kappa_1$ | proportion | (1.2, 5.5) | 1.24648 | [1.20212, 3.69887] | fitted |
| $\kappa_2$ | proportion | (1, 5) | 4.93806 | [1.42178, 4.99056] | fitted |
| $\delta_1$ | proportion | (0.0002, 0.001) | 0.000035 | [0.00020, 0.00099] | fitted |
| $\delta_2$ | proportion | (0.00003, 0.009) | 0.00264 | [0.00003, 0.00808] | fitted |
| $\gamma$ | week$^{-1}$ | (0.005, 0.075) | 0.00516 | [0.00505, 0.07441] | fitted |
| $\delta_r$ | proportion | (0.0001, 0.004) | 0.00031 | [0.00010, 0.00397] | fitted |

On the other hand, recovered female increases until week 30. On the other hand, exposed female starts decreasing after week three (Fig 4).

The asymptomatic and symptomatic VL cases attain peak close to week 20. Fig 5 depicts the total human VL infected cases. Again, most observation data points lie along the fitted curve or close to it. There is limitation in availability of sex-structured VL secondary data in literature. As a result, the sex-structured VL-data we have obtained from the literature have small sample (only five years data). To build confidence in the parameters estimated from the dynamical system from limited observational data, it is customary to apply bootstrapping

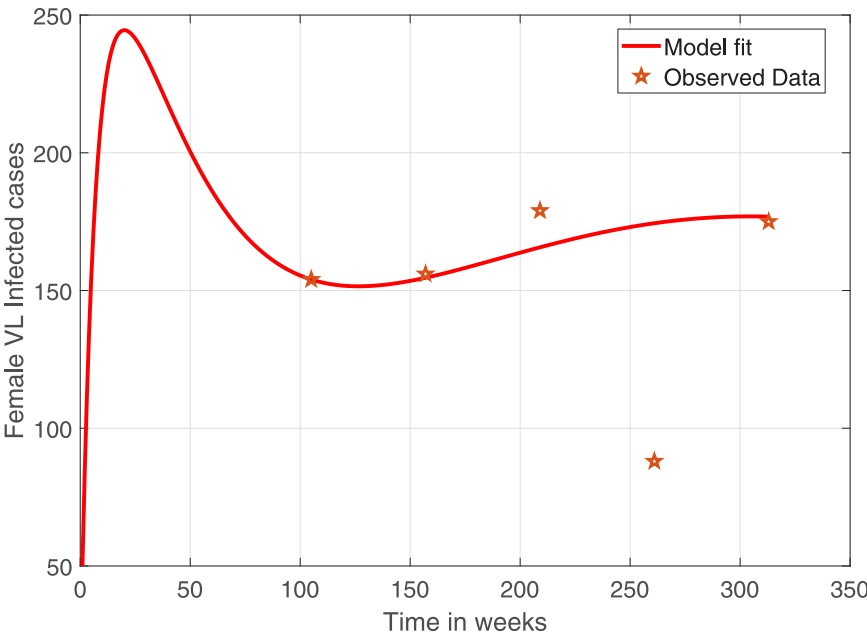

**Fig 2. Comparison of observed female human population VL confirmed cases in Ethiopia (indicated in stars) and model prediction (solid curve).** The starting week 0 stands for 2012 which is 2 years before the actual data. Parameter values used are as given in Table 4 and the initial conditions are as given above.

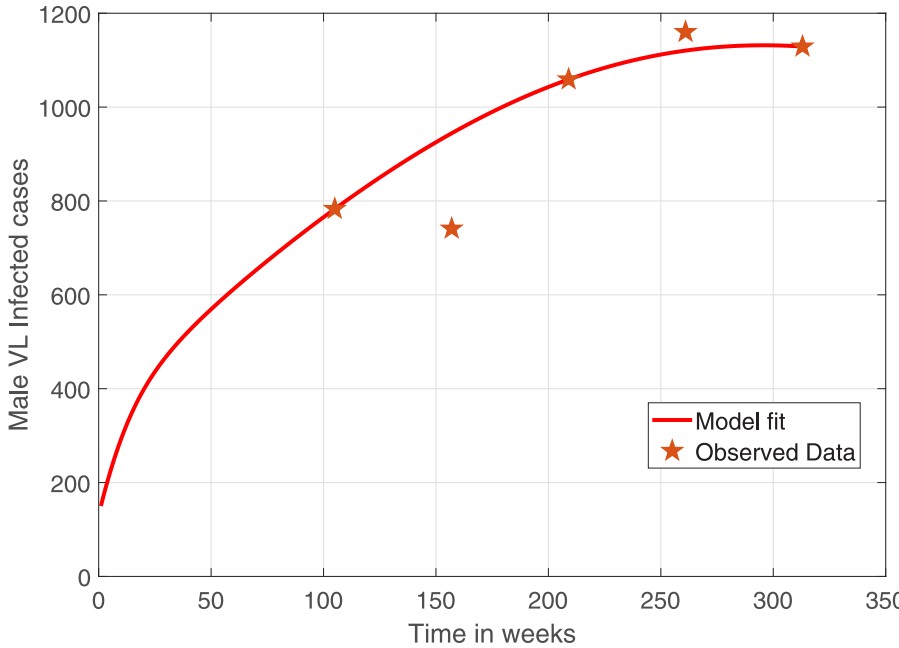

**Fig 3. Comparison of observed male human population VL confirmed cases in Ethiopia (indicated in stars) and model prediction (solid curve).** The starting week 0 stands for 2012 which is 2 years before the actual data. Parameter values used are as given in Table 4 and the initial conditions are as given above.

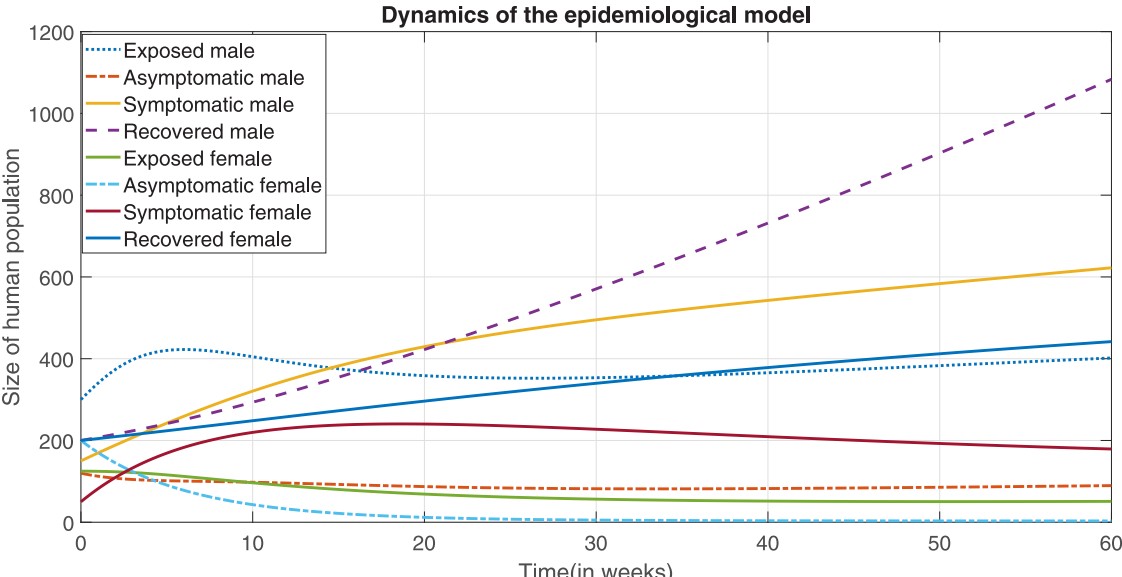

**Fig 4. The variation of the sizes of classes, exposed males and females, asymptomatic and symptomatic males and females, recovered males and females in time.**

techniques. Ordinary bootstrapping techniques may not work for dynamical system necessitating the use of improved bootstrapping techniques based on the residual obtained after initial fitting of the parameters [52]. The authors referred to this approach as residual based bootstrapping. The residual based method is summarized as follows:

Step 1: Fitting the actual data by optimizing the parameters.

Step 2: The difference between observation and model out using optimum parameter sets is taken as residual (error) separately for male and female population.

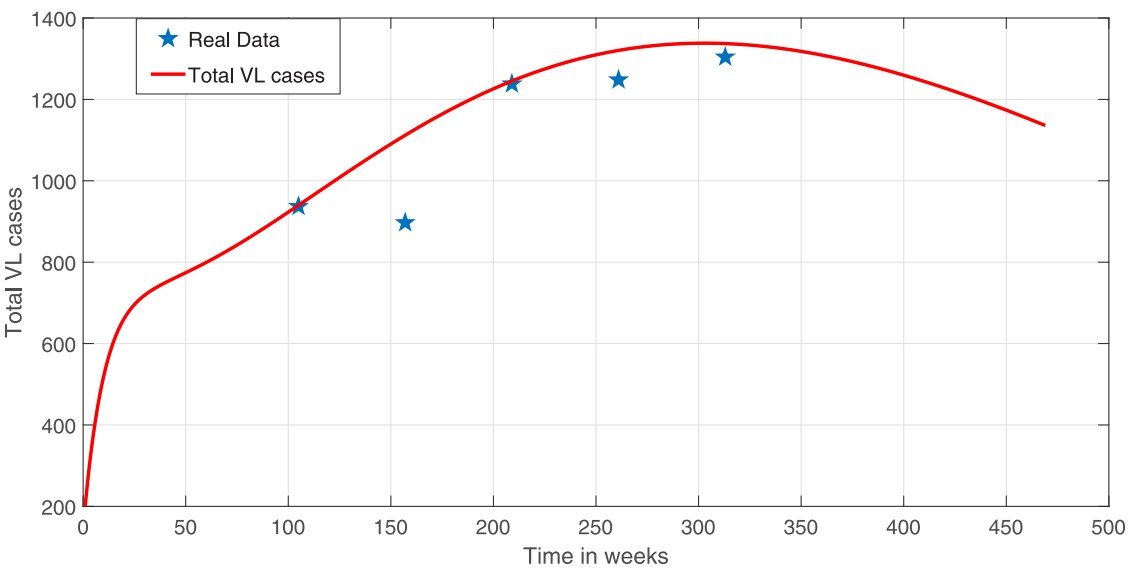

**Fig 5. Model fit to observed data corresponding to total VL infected cases the parameter values are as in Table 4.**

Step 3: The residual (error) is re-sampled 1000 times using bootstrapping techniques.

Step 4: The mean of this bootstrapped samples is added to the simulation output from Step 1 for the first time or Step 5 for subsequent runs to create a bootstrapped sample.

Step 5: The bootstrapped sample from Step 4 is fitted to the dynamical equations to obtain new optimum parameter sets.

Step 6: Steps 2 to 5 are repeated 100 times to generate 100 optimal estimates for each parameter.

The optimal estimates of each parameter obtained through the above procedure forms distribution of each estimated parameter from which confidence intervals can be drawn. From various methods proposed in literature to determine confidence intervals for parameters estimated from dynamic system model, we have chosen the percentile method. Using this method, the lower limit and upper limits of the estimated model parameters are the 5th percentile and the 95th percentile of the distribution of parameters. Table 4 shows the confidence intervals of the estimated model parameters.

## Sensitivity analysis

The VL model (1), developed in this study, contains 36 parameters. Hence, uncertainties are expected to arise in the estimates of the values of the parameters used in the numerical simulations of the model. Here, an uncertainty analysis, using Latin hypercube sampling (LHS), is carried out to account for the effect of such uncertainties on the numerical solutions. Furthermore, a global sensitivity analysis, using partial rank correlation coefficients (PRCC), is implemented to quantify the impact of the variations or sensitivity of each parameter of the model on the associated numerical simulations. The analysis are carried out for the sex structured VL model using the associated reproduction number as the response function (the parameter values and ranges in Table 4). It is assumed that the parameters of the model are uniformly distributed. Furthermore, 1000 LHS samples are generated.

Using the VL reproduction number ($\mathcal{R}_0$) as the response function, the results of the sensitivity analysis of the model (1), depicted in Fig 6, reveal that the top five PRCC-ranked parameters that play a more dominant role in the dynamics of VL are the recruitment rate of sand flies ($\Lambda_v$), the probability that a susceptible sand fly becomes infected when feed on an infected human ($\beta_{hv}$), the number of bites at which a male host receives per unit of time ($c_{1m}$), the number of times a single vector feeds on a male host ($c_{3m}$), and the probability that a susceptible reservoirs becomes infected through a single bite of an infected sand flies ($\beta_{rv}$). These parameters are positively correlated with the response function ($\mathcal{R}_0$). On the other hand, recovery rate of reservoirs ($\gamma$) and recruitment rate of reservoirs ($\Lambda_r$), with negative PRCCs have the greatest potential to sustain the disease in the community when increased. Thus, the public health implication of this result is that VL can be effectively controlled using by tuning these parameters towards the right direction to reduce VL reproduction number (Fig 7). For example, reducing the recruitment rate of vectors by using different vector control mechanisms such as insecticide-treated bed nets or insect repellents and reducing the contact rates of vectors and reservoirs have great impact to decrease the transmission of the disease. On the other hand, increasing the recovery rate $\gamma$ through treatment can possibly reduce the impact of the disease in the community. A pairwise comparison of the significant parameters (whose p -values are less than 0.05, see Table 5) is conducted to ascertain whether the processes described by the pair of parameters are different. The p-values for the different pairs of significant parameters is computed while accounting for the false discovery rate (FDR) as given in Table 6.

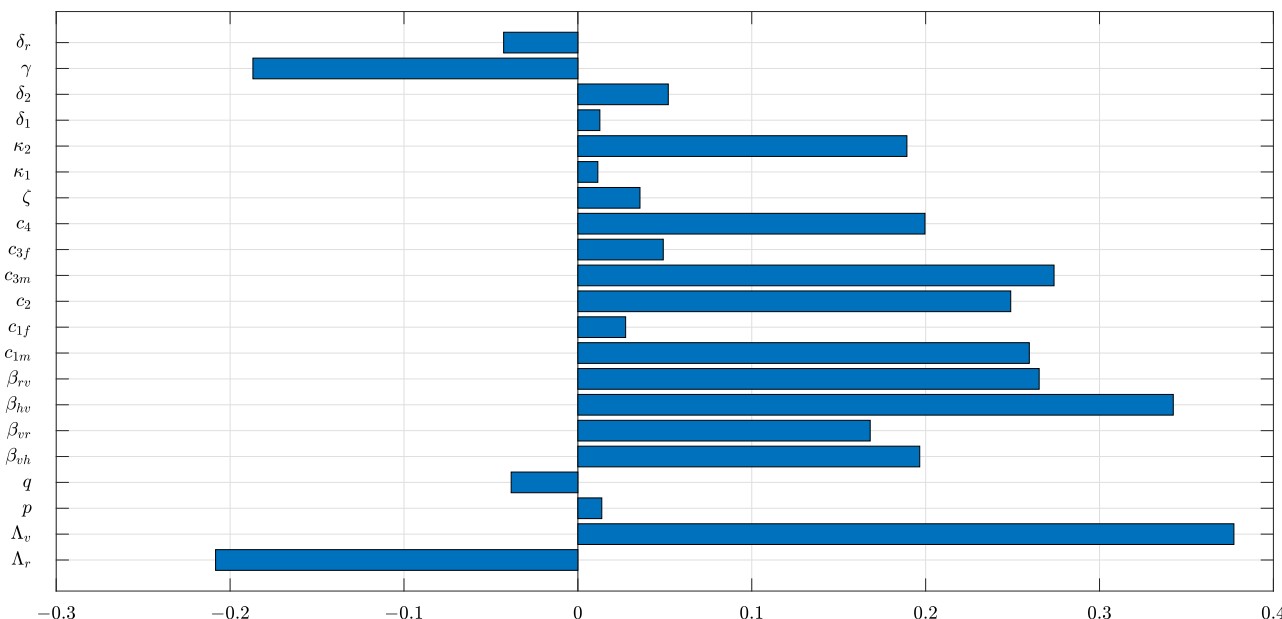

**Fig 6. Partial Rank Correlation Coefficients (PRCCs) for a selected range of model parameters in Table 4.** VL epidemic will be the worst when the parameters $\Lambda_v$, $\beta_{hv}$, $c_{1m}$, $\beta_{rv}$, and $c_{3m}$ are increased, whereas increasing $\Lambda_r$ and $\gamma$ will decrease the prevalence of disease.

The major question posed at this point is: Do the different pairs of significant parameters differ after FDR adjustment? Based on the FDR adjusted p-values in Table 6, the pair of parameters are rendered to be different if their p-value is less than 0.05 at 95% significance level and not different otherwise. We summarize the results in Table 7, where 'True' indicates that the

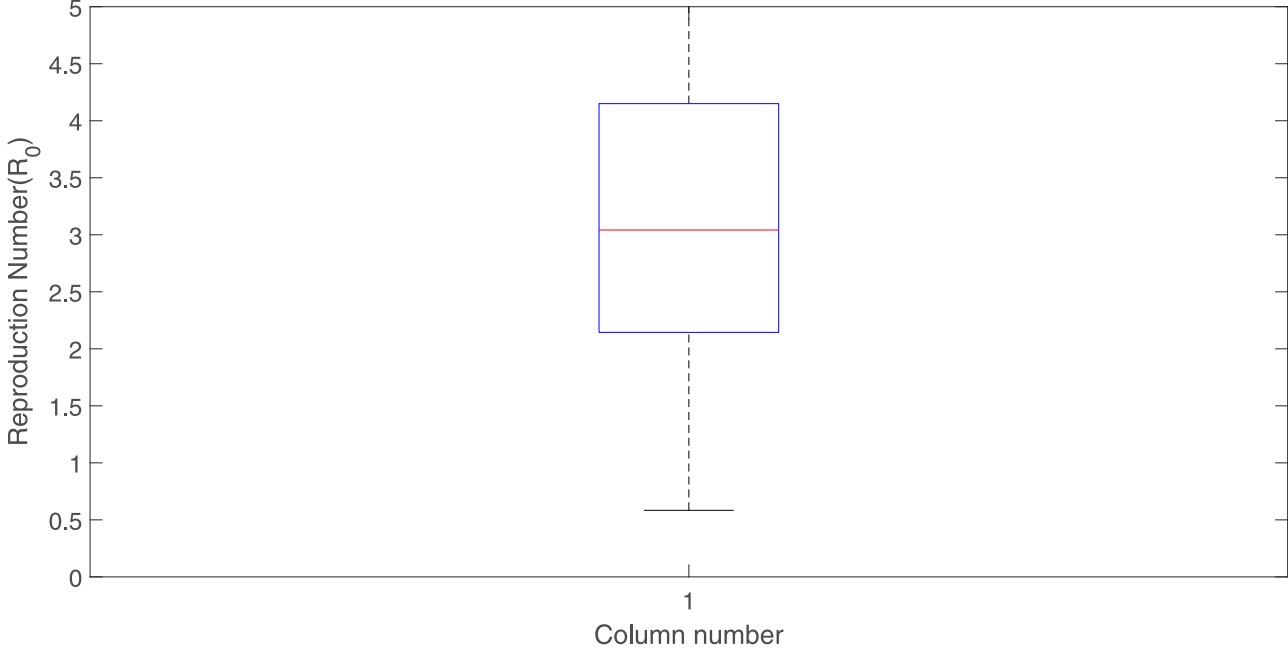

**Fig 7. The median of the basic reproduction number from 100 samples is 3.0419 (95% CI:2.1441–4.1501), the lower quartile of the computed value of the basic reproduction number ($\mathcal{R}_0$) is about 2.1441, and the upper quartile of 4.1501.**

**Table 5. The parameter PRCC significance (the False Discovery Rate (FDR) Adjusted p-value).**

| Variable | PRCC | p-value | Significance? |
|---|---|---|---|
| $\Lambda_r$ | -20835 | $1.15 \times 10^{-10}$ | True |
| $\Lambda_v$ | 0.37729 | 0.00000 | True |
| p | 0.01376082 | $7.196 \times 10^{-1}$ | False |
| q | -0.03835850 | $3.023 \times 10^{-1}$ | False |
| $\beta_{vh}$ | 0.19660 | $1.113 \times 10^{-09}$ | True |
| $\beta_{vr}$ | 0.16811 | $1.970 \times 10^{-07}$ | True |
| $\beta_{hv}$ | 0.34241 | 0.00000 | True |
| $\beta_{rv}$ | 0.26529 | 0.0000 | True |
| $c_{1m}$ | 0.29562933 | 0.0000 | True |
| $c_{1f}$ | 0.02747 | $4.554 \times 10^{-01}$ | False |
| $c_2$ | 0.24893 | $6.99 \times 10^{-15}$ | True |
| $c_{3m}$ | 0.273849 | 0.00000 | True |
| $c_{3f}$ | 0.04910 | $1.867 \times 10^{-01}$ | False |
| $c_4$ | 0.199632 | $6.654 \times 10^{-10}$ | True |
| $\zeta$ | 0.03572 | $3.261 \times 10^{-01}$ | False |
| $\kappa_1$ | 0.01148 | $7.196 \times 10^{-01}$ | False |
| $\kappa_2$ | 0.18923 | $4.488 \times 10^{-09}$ | True |
| $\delta_1$ | 0.01265 | $7.196 \times 10^{-01}$ | False |
| $\delta_2$ | 0.05199 | $1.677 \times 10^{-01}$ | False |
| $\gamma$ | -0.18687 | $6.503 \times 10^{-09}$ | True |
| $\delta_r$ | -0.04268 | $2.546 \times 10^{-01}$ | False |

**Table 6. The pairwise PRCC comparisons (FDR Adjusted P-values).**

| | $\Lambda_r$ | $\Lambda_v$ | $\beta_{vh}$ | $\beta_{vr}$ | $\beta_{hv}$ | $\beta_{rv}$ | $c_{1m}$ |
|---|---|---|---|---|---|---|---|
| $\Lambda_r$ | | 0 | 0 | 0 | 0 | 0 | 0 |
| $\Lambda_v$ | | | $3.573 \times 10^{-06}$ | $1.616 \times 10^{-06}$ | 0.4679 | 0.01211 | 0.008217 |
| $\beta_{vh}$ | | | | 0.6178 | 0.001209 | 0.1558 | 0.1947 |
| $\beta_{vr}$ | | | | | $9.364 \times 10^{-05}$ | 0.04541 | 0.06215 |
| $\beta_{hv}$ | | | | | | 0.1017 | 0.07852 |
| $\beta_{rv}$ | | | | | | | 0.9066 |

**Table 7. Are the parameters different after FDR adjustment?.**

| | $\Lambda_r$ | $\Lambda_v$ | $\beta_{vh}$ | $\beta_{vr}$ | $\beta_{hv}$ | $\beta_{rv}$ | $c_{1m}$ | $c_2$ | $c_{3m}$ | $c_4$ |
|---|---|---|---|---|---|---|---|---|---|---|
| $\Lambda_r$ | | TRUE | TRUE | TRUE | TRUE | TRUE | TRUE | TRUE | TRUE | TRUE |
| $\Lambda_V$ | | | TRUE | TRUE | FALSE | TRUE | TRUE | TRUE | TRUE | TRUE |
| $\beta_{vh}$ | | | | FALSE | TRUE | FALSE | FALSE | FALSE | FALSE | FALSE |
| $\beta_{vr}$ | | | | | TRUE | TRUE | FALSE | FALSE | TRUE | FALSE |
| $\beta_{hv}$ | | | | | | FALSE | FALSE | TRUE | FALSE | TRUE |
| $\beta_{rv}$ | | | | | | | FALSE | FALSE | FALSE | FALSE |
| $c_{1m}$ | | | | | | | | FALSE | FALSE | FALSE |
| $c_2$ | | | | | | | | | FALSE | FALSE |
| $c_{3m}$ | | | | | | | | | | FALSE |

compared parameters are significantly different and 'False' indicating otherwise. We observe that some sensitive parameters are not significantly different. For example, $\Lambda_v - \beta_{hv}$, $\beta_{vh} - \beta_{vr}$, $\beta_{vh} - \beta_{rv}$, $\beta_{vh} - c_{1m}$, $\beta_{vh} - c_{3m}$, $\beta_{vr} - c_{1m}$, $\beta_{hv} - \beta_{rv}$, $\beta_{hv} - c_{1m}$, $\beta_{hv} - c_{3m}$ are the most sensitive parameters but the pairs are not different (see Table 7).

## Numerical simulation and discussion

There are various intervention strategies for Visceral Leishmaniasis that are being implemented in different endemic areas of the world. These mitigation strategies are classified into three main categories:

1. Personal protection and environmental hygiene. Applying clothing that covers as much of the body as possible, sleeping on higher floors because the vectors can't fly long distances, avoiding being outside between dusk and dawn, using insecticide-treated nets [54].

2. Treatment by the use of WHO recommended VL therapies are recommended despite the fact that VL therapy is a difficult task to manage because of its variable effectiveness, side effects, and cost. It is treatable and curable provided that treatment is started right away and is effectively completed. Currently, the WHO recommended and available medications for VL are liposomal amphotericin B, Miltefosine, Pentavalent antimonials (such as sodium stibogluconate (SSG)), and Paromomycin [55].

3. Vector control (Sand fly control) requires systematic destruction of their habitat and the use of indoor and outdoors spraying of insecticides, residual treatment of human dwellings and/or animal shelters' walls, treating tree trunks and vegetation.

To investigate the outcomes of VL control strategies stated above, we used parameters that represent additional efforts (1) in preventive controls for female and male humans, (2) for sand fly control, and (3) for treatment of humans and reservoirs in the model system (1). The next section considers the implementation of different VL control strategies on each sex separately.

**Interventions on females.** Implementing the VL prevention strategies such as using insecticide-treated bed nets, wearing clothing that covers as much skin as possible, sleeping on higher floors, avoiding being outside between dusk and dawn could help to reduce the contact rate with vectors and consequently the sand fly biting rate. Furthermore, insecticide-treated bed net kills vectors thereby increasing average vectors death rate. We have simulated the model system (1) starting from 0 case of VL in the year 2012 which is two years before our first recorded VL data. The intervention started at the last VL recorded data in 2018 and is assumed to continue for the next 157 weeks as shown in Fig 8.

The interventions to reduce the number of sand fly bites through various strategies mentioned above is parameterized as $c_{1f}$ in model system (1). Therefore, various scale of intervention can be represented by different values of $c_{1f}$. In Fig 8 a 10%, 30% and 60% reduction in $c_{1f}$ from a baseline value led to a decrease in VL cases by above 209.37, 240.92 and 288.13 cases after 157 weeks of intervention. The reduction by 16.11%, 18.53% and 22.16% of the total VL cases of 1300 at the beginning of the intervention is modest but important step forward VL transmission control along with other measures such as anti-leishmaial therapies.

This suggests that while an additional 60% effort to shield female human populations from sand fly bites could reduce the overall number of VL cases by about 22%, this intervention would not be sufficient to manage the disease within the community. Therefore, another intervention has to be considered together with this.

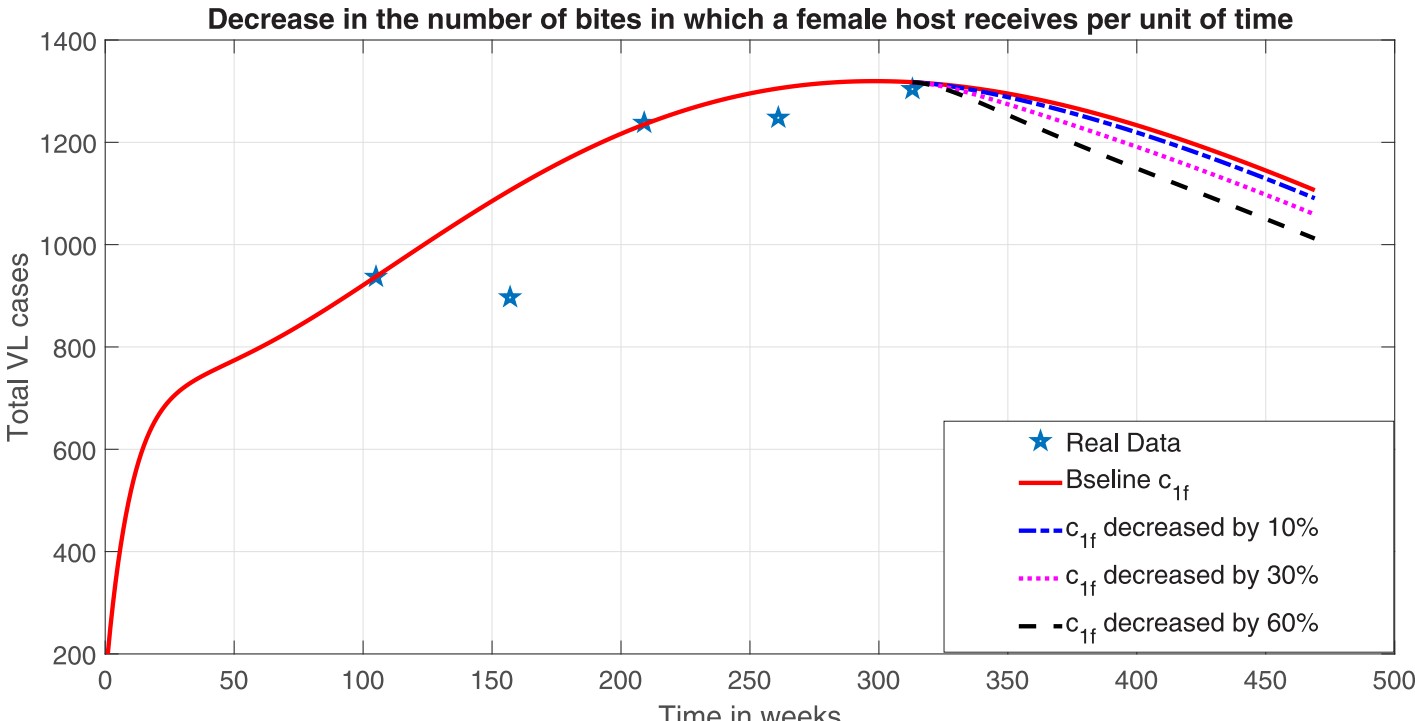

**Fig 8. Effect of reducing the biting rate in which female human host receives per week by a given amount from the baseline value, where the other parameter values are as in Table 4.**

The anti-leishmanial treatment approach depends in part by host and parasite factors as well as specific section of the population, such as children, the elderly, pregnant women, immunocompromised individuals, and those with additional comorbidity. In our model system (1), the recovery rate of female human population ($\tau_2$) is updated by $(1 + r)\tau_2$, where $r$ is an additional effort that is applied on recruiting infected females to get treated with VL drugs. When r is increased by 50% (i.e if 50% additional treatment effort is applied) only on female patients under full efficacy of the drugs, our numerical simulation result (Fig 9) shows that, the total VL prevalence will decrease by 18.97%. This is a hypothetical intervention as suggested by the model system (1). However, actual implementation should take a number of factors into considerations to achieve the same level of VL reduction as in the model. For example, it is crucial to counsel vulnerable female human populations to utilize VL preventative measures and female VL patients to effectively complete their treatment course by consulting with their physician in order to reduce the total prevalence of the disease.

**Interventions on males.** Men are more frequently affected by VL disease than women as noted in the previousc sections. Applying protective mechanisms mainly on men hosts will bring a significant decrease in the total number of VL cases. Therefore, it is important to design VL control strategies that will decrease the number of VL-infected men significantly. Similar to the application of additional efforts on females, we also applied the same procedure for males to simulate the impact on the number of VL cases. The simulation with a reduced sand fly biting rate that a male host receives, $c_{1m}$, from base rate by 10%, 30% and 60% led to a reduction of about 30.89%, 56.04% and 80% respectively after 157 weeks of interventions (Fig 10).

In contrast to the interventions on female, the interventions on male population has led significant reduction clearly showing the significant difference between sexes on VL

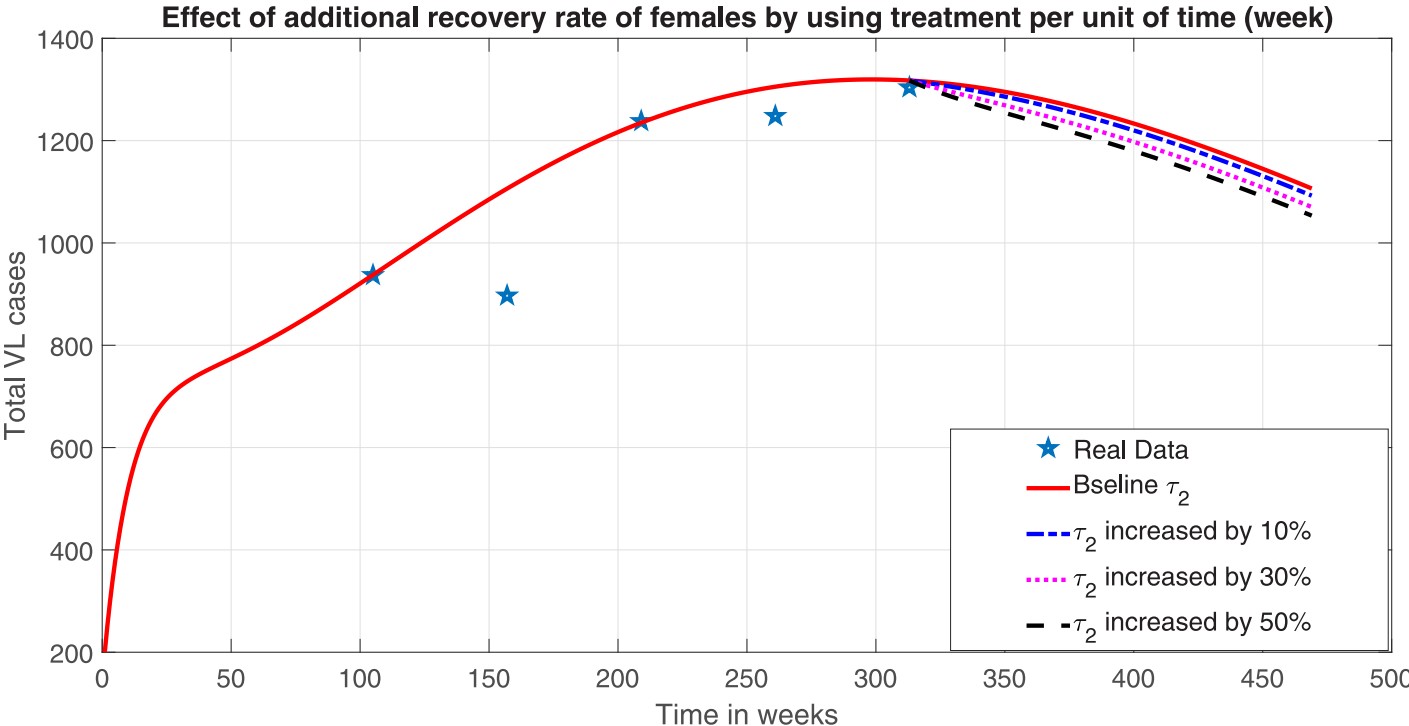

**Fig 9. Effect of additional effective treatments only for female.**

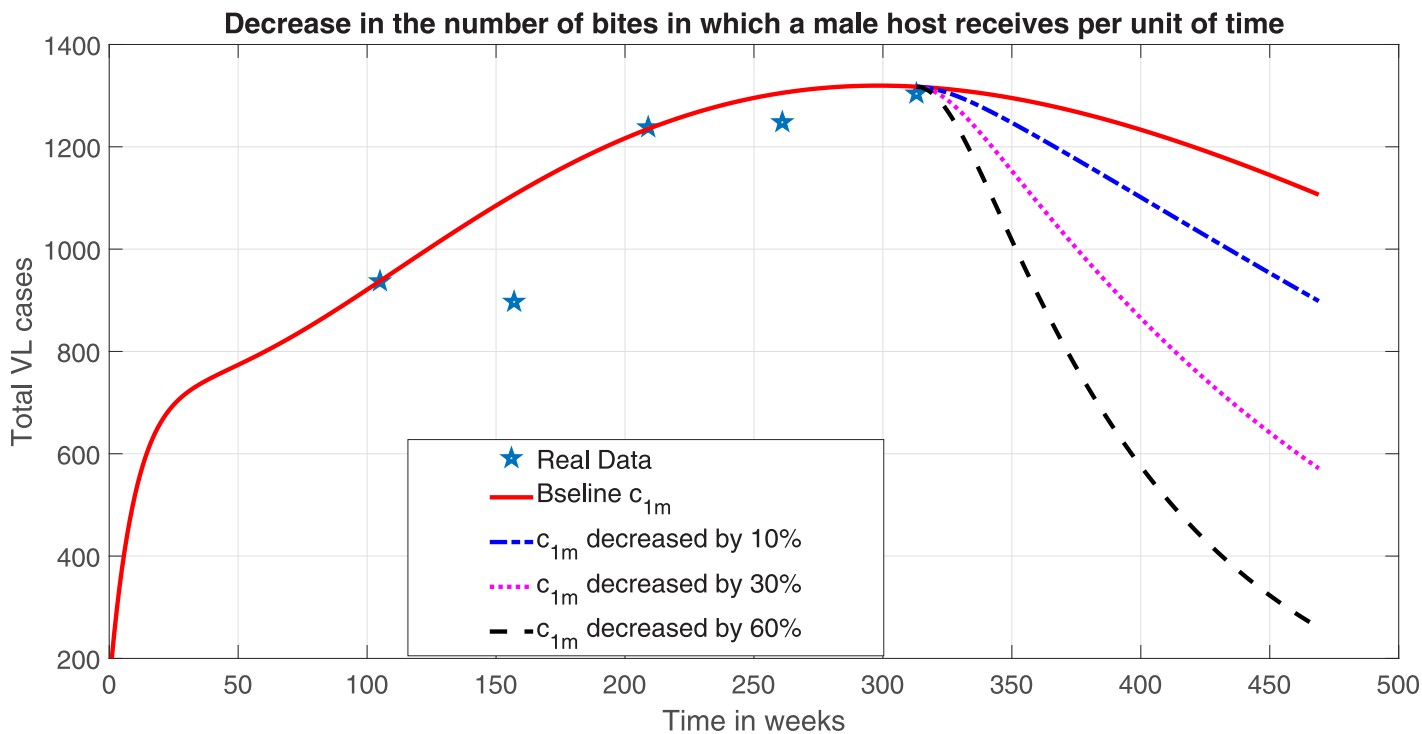

**Fig 10. Effect of reducing the number of bites in which a male human host receives per week by the same percentage from the baseline value.**

transmission. The hypothetical model simulations can be realized by protecting males from the bite of sand flies through the use of insecticide-treated bed-nets, wearing clothes that cover as much as skin as possible, avoiding movement during the evening (when sand flies are active), and reducing travels into endemic areas. Therefore, all available preventive measures beyond the above list must be used to shied males in larger proportion in order to contain the disease.

The other VL control mechanism is the use of effective drugs. In our model system (1), the recovery rate of male human population ($\tau_1$) is updated by $(1 + t)\tau_1$, where $t$ is an additional percentage effort that is applied on infected males to use VL drugs. When a 50% additional treatment effort is applied on infected males under full efficacy of the drugs, our numerical simulation result (Fig 11) shows that, the total VL prevalence will decrease by 69.5% (Fig 11).

Therefore, in addition to providing efficient medical treatment for infected males, all possible preventative measures must be implemented in larger proportion to protect susceptible men against sand fly bites in order to eradicate the disease from the community. It is evident that combining both intervention options at the same time can lead to a significant decrease in VL cases from the male host even at lower values of additional efforts to reduce infection and enhance treatment.

**Vector control.**   Since vector-borne diseases such as visceral leishmaniasis can have fatal outcomes if left untreated, effective vector management is a crucial public health strategy for controlling the disease. The use of insecticides and managing vector breeding sites are the major strategies for vector control. Vector control can be administered either by serious destruction of their breeding sites that help to kill the vector at their pre-adult stages or by the use of insecticide-treated bed nets and indoor/outdoor residual spraying. Implementation of these control mechanisms reduces the recruitment rate of vectors. Let $\alpha$ represent the rate at

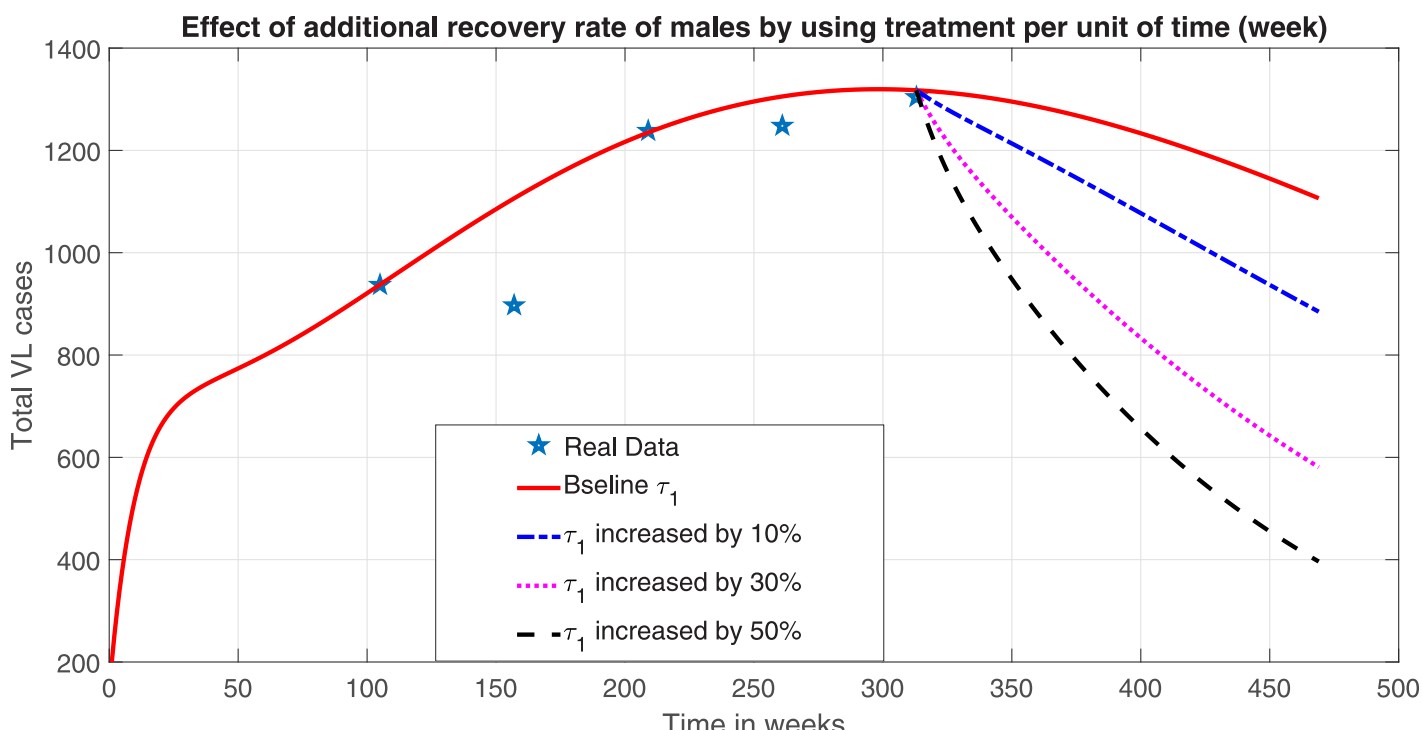

**Fig 11.  Effect of additional efforts in recruitment of males human host for effective treatment.**

which the effort reduces the proportion of the vector on the pre-adult stage. Then the remaining proportion $(1 - \alpha)\Lambda_v$ of vectors survive and become parts of the active adult sand fly population. In addition to this, application of insecticides spray can possibly kill the adult sand fly. Let $\rho$ represent the additional effort that is applied to kill adult sand flies per week. Then, the natural death rate of vectors ($\mu_v$) will be updated as $\mu_v + \rho$. Due to the difficulty of managing vectors, we considered the maximum additional effort of using insecticide treated bed net as well as that of insecticide spray to be 7.5%. When we include this formulation into the model system (1) and integrate the system, we got numerical solution depicted in Fig 12.

Our numerical result shows that when the vector control strategies are used and the number of vectors are reduced by up to 15% from the baseline value, the total VL prevalence reduces to 57.71%. This shows that vector control strategy is highly sensitive and can be used to significantly reduce the overall prevalence of the disease and subsequently control the transmission.

By implementing the use of insecticide-treated bed nets and insecticide spray, the number of vectors can be reduced, which will greatly lower the overall prevalence of the disease and help in its containment within the community. This disease control mechanism is crucial like that of protecting males from the bite of the sand flies.

**Interventions on reservoirs.** Dogs have been identified as reservoir hosts for Leishmania, and studies indicate that domestic, and wild mammals, such as foxes, pigs, cats, and rock hyraxes, can be infected by the VL [56]. Reducing the number of wild rodents, feral dogs, and other mammalian hosts from the region inhabited by humans, as well as treating and screening domestic dogs, are crucial steps in managing reservoirs as one of the control strategies of VL transmission. Hence, in model system (1), an additional percentage of full recovery rate of reservoirs is represented by $v$%. Then the recovery rate ($\gamma$) is updated by $(1 + v)\gamma$ in the model

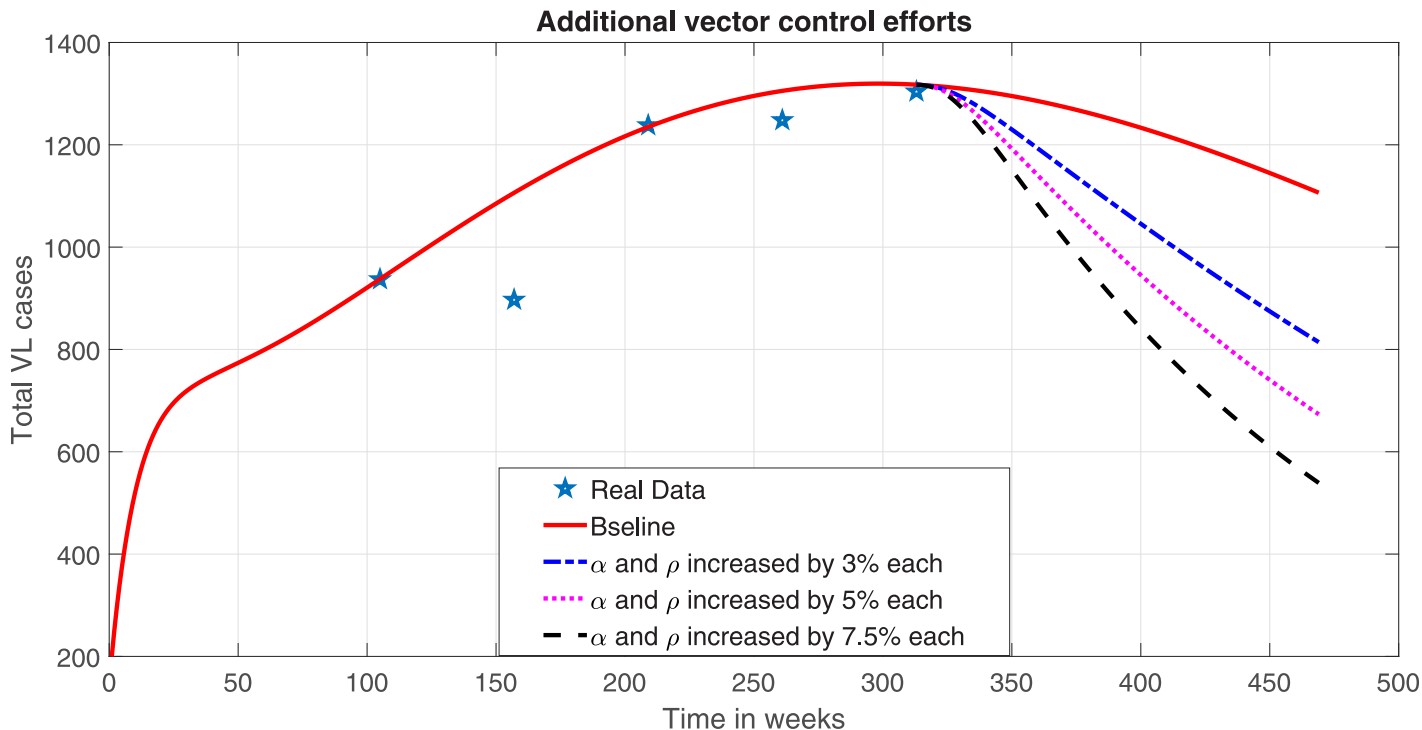

**Fig 12. Effect of vector control.**

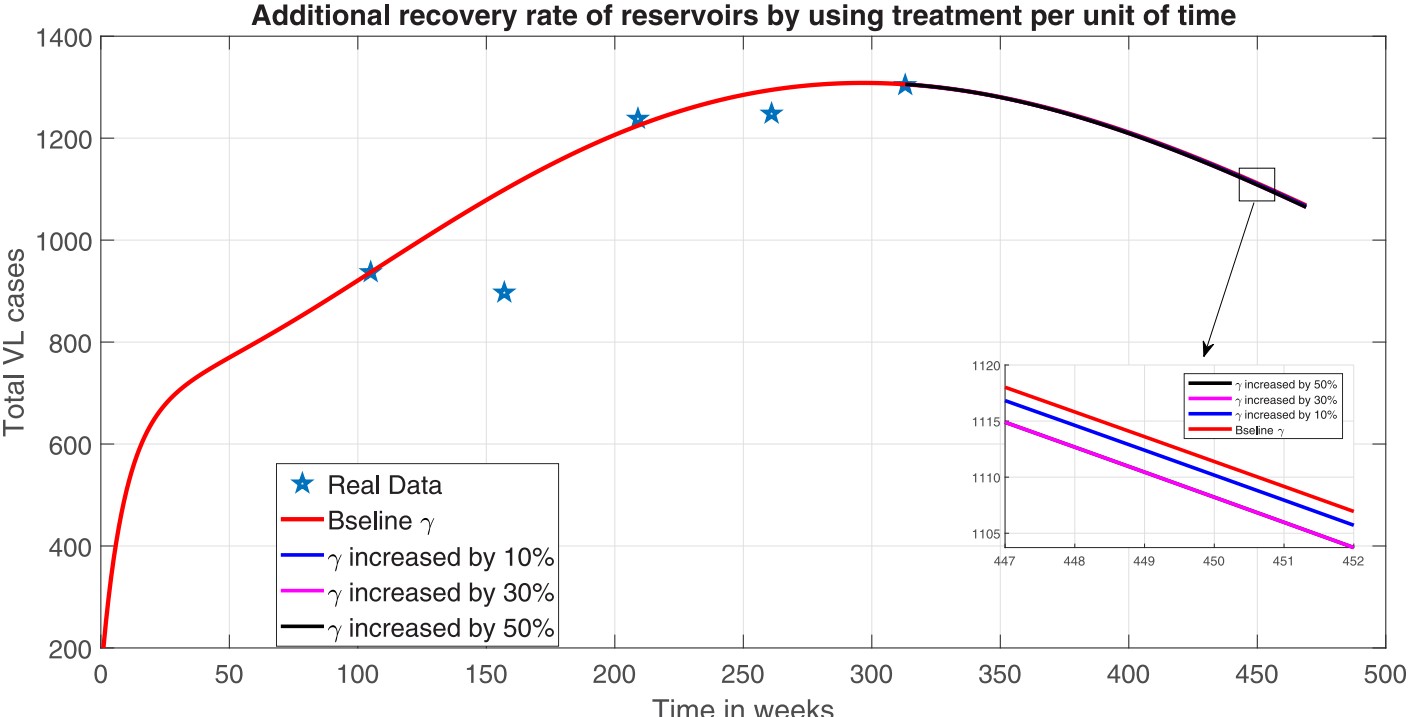

**Fig 13. Effect of reservoir treatment.**

system (1). We used the maximum full recovery rate of 50% to see the impact on VL prevalence. However, the numerical simulation shows that, treatment of reservoirs have no significant contribution in the reduction of total prevalence in humans (Fig 13). But, close inspection of Fig 13 shows that this intervention also contributes to the decrease in the total prevalence of the disease in humans but at a much slow rates.

## Conclusions and future works

Leishmaniasis are a group of diseases caused by more than 20 species of the protozoan genus Leishmania that are transmitted between humans and other mammalian hosts by the vector sand flies. The disease is prevalent in many countries globally affecting an estimated 12 million people with an additional 350 million at risk of contracting with significant portion of these figures in Africa and Asia. As a result, a number of past studies focused on developing effective disease control strategies for susceptible population with no distinction between sexes. However, the disease affects more male than female. Therefore, most of the proposed interventions were not effective. This study proposed new model where male and female interventions and treatments are separately considered. The vector and reservoir control strategies are also separately considered forming what is referred as compartmental VL model in this study.

Qualitative analysis of the model positivity, boundedness, stability of equilibrium points and backward bifurcation have been discussed. The analysis of the model shows that, there is a possibility for a backward bifurcation for $\mathcal{R}_0$. This phenomenon has an epidemiological consequence whereby the traditional epidemiological requirement of lowering (and maintaining) $\mathcal{R}_0$ to a value less than unity, while still necessary, is no longer sufficient for the effective control (or elimination) of the disease.

The VL model developed in this study has many model parameters. As a result, it is anticipated that there will be uncertainty in the estimations of the values of the parameters employed in the model in views of limited clinical data on VL. Latin hypercube sampling (LHS)- based uncertainty analysis is conducted to evaluate the impact of such uncertainties on the outcomes of the numerical simulation. The basic reproduction number is used as the response function in the analysis of the sex-structured VL model. Amhara's region VL data from Ethiopia are used to estimate the parameters. It is shown that, the lower quartile of the computed values of $\mathcal{R}_0$ is about 2.1441, the median around 3.0419 and the upper quartile of about 4.1501. Using the VL reproduction number as the response function, the results of the sensitivity analysis of the model reveal that the top four PRCC-ranked parameters that play a more dominant role in the dynamics of VL are the recruitment rate of sand flies, the number of bites at which a male host receives per unit of time, the transmission probability of human, and the probability that a susceptible sand fly becomes infected when feeding on an infected human. These parameters are found to be positively correlated with the response function ($\mathcal{R}_0$). On the other hand, the recovery rate of reservoirs and recruitment rate of reservoirs, with negative PRCCs have the greatest potential to sustain the disease in the community when they are increased. The public health implication of this result is that VL can be effectively controlled using strategies that significantly reduce these parameters.

The impact of sex in controlling the transmission of visceral leishmaniasis was assessed. Reducing the contact rate of males with vectors through various interventions is found to decrease the prevalence of the disease at a much faster than other interventions leading to elimination of the disease. Vector control is found to be another effective tool of containing the disease from the community. Furthermore, use of effective medical treatment on infected male is also an important disease control mechanism. Application of preventive and medical treatment on female human population shouldn't be ignored even though its impact to control the disease is not as significant as males. Since sand fly is not a long distance flying vector, keeping the distance of reservoirs from human population will decrease the biting rate by the vector. The numerical solution of the model show that male human population are the important agent in decreasing the prevalence of the disease.

Our numerical result also reveals that the implementation of disease-preventive strategies, such as utilizing insecticide-treated bed nets to reduce the biting rate that male human hosts receive, can significantly reduce the prevalence of the disease. Furthermore, the implementation of vector management strategies, such as eliminating pre-adult stage vectors and applying indoor and outdoor spraying, has been shown to considerably reduce the overall prevalence of the disease. When the male human host and reservoir host receive effective medical care, the prevalence will decrease. But, it is demonstrated that additional medical treatment efforts on reservoir animals have no significant effect on the overall prevalence of the disease as compared to other possible mechanisms. The numerical simulation infers that applying a maximum of 15% additional effort to reduce the number of vectors, decreases the total VL prevalence by 57.71%. It is also possible to decrease the total prevalence of VL by 69.5% when up to 50% more infected males receive treatment with 100% efficacy. Moreover, a maximum of 60% of extra preventative measures only on men humans considerably reduces the total prevalence of VL by 80%. To reduce the disease burden of visceral leishmaniasis, public health officials and concerned stakeholders need to give more emphasis to the proportion of male humans in their intervention strategies.

Therefore, in order to eradicate visceral leishmaniasis from a community, public health officials and concerned stakeholders must take into account the sensitivity of the different parameters of the compartmental model when designing actual preventive and treatment tools. In particular, there is a clear need for differential preventive and treatment strategies as revealed

in this study. It is necessary to note that the relative sensitivity of the model parameters are the only information needed to develop actual control mechanism from the hypothetical simulation in this study. The optimal combination of different tools is beyond the scope of the current study.

Financing and resources for healthcare are limited, and some treatments are highly expensive. Cost-effectiveness analysis is a useful tool for determining the value of healthcare interventions in relation to their costs. Therefore, it would be important to investigate in future works the cost-effectiveness of different strategies together with sex-structured visceral leishmaniasis model. Moreover, the abundance of vectors and correspondingly their contacts with hosts vary from season to season. Hence, it is also interesting to consider in the future work that the effect of seasonal variation on VL transmission in contrast to the sex lines.

## Acknowledgments

The first author would like to thank ORTARChI initiatives and Simons Foundation based in Botswana International University of Science and Technology (BIUST).

## Author Contributions

**Conceptualization:** Temesgen Debas Awoke, Semu Mitiku Kassa.

**Data curation:** Temesgen Debas Awoke.

**Formal analysis:** Temesgen Debas Awoke.

**Funding acquisition:** Gizaw Mengistu Tsidu.

**Investigation:** Temesgen Debas Awoke, Kgomotso Suzan Morupisi.

**Methodology:** Temesgen Debas Awoke, Semu Mitiku Kassa.

**Project administration:** Semu Mitiku Kassa, Gizaw Mengistu Tsidu.

**Software:** Temesgen Debas Awoke.

**Supervision:** Semu Mitiku Kassa, Kgomotso Suzan Morupisi, Gizaw Mengistu Tsidu.

**Validation:** Temesgen Debas Awoke, Gizaw Mengistu Tsidu.

**Visualization:** Temesgen Debas Awoke.

**Writing – original draft:** Temesgen Debas Awoke.

**Writing – review & editing:** Temesgen Debas Awoke, Semu Mitiku Kassa, Kgomotso Suzan Morupisi, Gizaw Mengistu Tsidu.

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
