## [Decision Letter · Decision Letter 0]

8 Jan 2024

PONE-D-23-39710Sex-structured disease transmission model and control mechanisms for Visceral leishmaniasis (VL)PLOS ONE

Dear Dr. Awoke,

Thank you for submitting your manuscript to PLOS ONE. After careful consideration, we feel that it has merit but does not fully meet PLOS ONE’s publication criteria as it currently stands. Therefore, we invite you to submit a revised version of the manuscript that addresses the points raised during the review process.

**The reviewer raises concerns about parameter fitting and I agree. While the authors performed sensitivity analysis, the model is prone to overfitting due to so many parameters and so little data points. **

**As the reviewer suggest, please provide confidence intervals. Also, for as many parameters as possible, look for values in the medical/biological literature. For example, is the loss of immunity (psir and psi h) or the incubation period (eta and phi) truly so different for males and females? Or is it just a consequence of parameter fitting? To me, such a difference sounds unrealistic. At the same time I am confident that even if you set those parameters to any reasonable number you find in the literature and then fit the remaining parameters to your data, you will still see a good fit. **

We look forward to receiving your revised manuscript.

Kind regards,

Jan Rychtář

Academic Editor

PLOS ONE

Journal Requirements:

This work was carried out with the aid of a grant from the O.R. Tambo Africa Research Chairs Initiative as supported by the Botswana International University of Science and Technology, the Ministry of Tertiary Education, Science and

Technology; the National Research Foundation of South Africa (NRF); the Department of Science and Innovation of South Africa (DSI); the International Development Research Centre of Canada (IDRC); and the Oliver & Adelaide Tambo Foundation (OATF). 

The authors would like to acknowledge that this work was carried out with the aid of a grant from the O.R. Tambo Africa Research Chairs Initiative as supported by the Botswana International University of Science and Technology, the Ministry of Tertiary Education, Science and Technology; the National Research Foundation of South Africa (NRF); the Department of Science and Innovation of South Africa (DSI); the International Development Research Centre of Canada (IDRC); and the Oliver & Adelaide Tambo Foundation (OATF). The first author also gratefully acknowledges the funding he received from Simons Foundation, based in Botswana International University of Science and Technology (BIUST).

This work was carried out with the aid of a grant from the O.R. Tambo Africa Research Chairs Initiative as supported by the Botswana International University of Science and Technology, the Ministry of Tertiary Education, Science and

Technology; the National Research Foundation of South Africa (NRF); the Department of Science and Innovation of South Africa (DSI); the International Development Research Centre of Canada (IDRC); and the Oliver & Adelaide Tambo Foundation (OATF). 

6. We note that your Data Availability Statement is currently as follows: All relevant data are within the manuscript and its Supporting Information files

Additional Editor Comments:

The reviewer raises concerns about parameter fitting and I agree. While the authors performed sensitivity analysis, the model is prone to overfitting due to so many parameters and so little data points.

As the reviewer suggest, please provide confidence intervals. Also, for as many parameters as possible, look for values in the medical/biological literature. For example, is the loss of immunity (psir and psi h) or the incubation period (eta and phi) truly so different for males and females? Or is it just a consequence of parameter fitting? To me, such a difference sounds unrealistic. At the same time I am confident that even if you set those parameters to any reasonable number you find in the literature and then fit the remaining parameters to your data, you will still see a good fit.

Reviewers' comments:

Reviewer's Responses to Questions

**Comments to the Author**

1. Is the manuscript technically sound, and do the data support the conclusions?

Reviewer #1: Yes

2. Has the statistical analysis been performed appropriately and rigorously? 

Reviewer #1: No

3. Have the authors made all data underlying the findings in their manuscript fully available?

Reviewer #1: No

4. Is the manuscript presented in an intelligible fashion and written in standard English?

Reviewer #1: Yes

5. Review Comments to the Author

Reviewer #1: I have read the paper and overall the paper is ok but there are major things that the authors need to consider

1. I am wondering about the fitted values. WIth the small number of data and fit many parameters, I think it is hard to get the appropriate results. Please clarify with confidence interval

2. THe numerical solutions are unclear. PLease provide explanation for each figures not only state the results but the implication of the results

6. PLOS authors have the option to publish the peer review history of their article (what does this mean?). If published, this will include your full peer review and any attached files.

Reviewer #1: No

---

## [Author Response · Author response to Decision Letter 0]

23 Feb 2024

All comments are incorporated in the revised manuscript. Regarding data availability, since we used secondary data and already cited the source, we have no supporting information file. All the data are in the manuscript. Apology for the inconvenience.

---

## [Editor Report · Decision Letter 1]

12 Mar 2024

Sex-structured disease transmission model and control mechanisms for Visceral leishmaniasis (VL)

PONE-D-23-39710R1

Dear Dr. Awoke,

We’re pleased to inform you that your manuscript has been judged scientifically suitable for publication and will be formally accepted for publication once it meets all outstanding technical requirements.

Kind regards,

Jan Rychtář

Academic Editor

PLOS ONE
---

## [Editor Report · Acceptance letter]

17 Mar 2024

PONE-D-23-39710R1 

PLOS ONE

Dear Dr. Awoke, 

I'm pleased to inform you that your manuscript has been deemed suitable for publication in PLOS ONE. Congratulations! Your manuscript is now being handed over to our production team.

Kind regards, 

on behalf of

Dr. Jan Rychtář 

Academic Editor

PLOS ONE